# Differences in regional brain structure in toddlers with autism are related to future language outcomes

Kuaikuai Duan [1] ✉, Lisa Eyler[2,3], Karen Pierce[1], Michael V. Lombardo [4], Michael Datko[1], Donald J. Hagler[5], Vani Taluja [1], Javad Zahiri[1], Kathleen Campbell [1], Cynthia Carter Barnes[1], Steven Arias[1], Srinivasa Nalabolu[1], Jaden Troxel[1], Peng Ji[6] & Eric Courchesne [1] ✉

Language and social symptoms improve with age in some autistic toddlers, but not in others, and such outcome differences are not clearly predictable from clinical scores alone. Here we aim to identify early-age brain alterations in autism that are prognostic of future language ability. Leveraging 372 longitudinal structural MRI scans from 166 autistic toddlers and 109 typical toddlers and controlling for brain size, we find that, compared to typical toddlers, autistic toddlers show differentially larger or thicker temporal and fusiform regions; smaller or thinner inferior frontal lobe and midline structures; larger callosal subregion volume; and smaller cerebellum. Most differences are replicated in an independent cohort of 75 toddlers. These brain alterations improve accuracy for predicting language outcome at 6-month follow-up beyond intake clinical and demographic variables. Temporal, fusiform, and inferior frontal alterations are related to autism symptom severity and cognitive impairments at early intake ages. Among autistic toddlers, brain alterations in social, language and face processing areas enhance the prediction of the child's future language ability.

Autism spectrum disorder (ASD) is a neurodevelopmental disorder characterized by social and communicative deficits and repetitive behaviors emerging at ages 1 to 4 years[1,2]. ASD affects ~1 in 44 children in the United States[3]. The high prevalence rate of ASD and associated social and language deficits significantly elevate the risk of adverse outcomes for individuals with ASD and increase the burden for the involved families and the whole society. Clinical heterogeneity of ASD is considerable[4–8]: Some toddlers benefit from contemporary applied behavior analysis treatments[9], but others do not[10,11]. Some toddlers may go on to earn college degrees and live independently, but others

remain minimally-verbal with life-long struggles with social communication. While language and social symptoms improve with age in some toddlers, they do not for others, and such outcome differences are not clearly predictable from clinical scores at very early ages[4–8]. Characterizing ASD neuropathology at the age of clinical onset, and how it relates to clinical heterogeneity, is essential for aiding early diagnosis, prognosis, and early interventions.

Converging evidence from neuroanatomical studies suggests brain or cortical overgrowth in young children with ASD[12–19], especially in frontal and temporal regions[2,18–21]. ASD toddler-derived brain

[1]Autism Center of Excellence, Department of Neurosciences, University of California, San Diego, La Jolla, CA 92037, USA. [2]Department of Psychiatry, University of California, San Diego, La Jolla, CA 92093, USA. [3]VISN 22 Mental Illness Research, Education, and Clinical Center, VA San Diego Healthcare System, San Diego, CA 92161, USA. [4]Laboratory for Autism and Neurodevelopmental Disorders, Center for Neuroscience and Cognitive Systems @UniTn, Istituto Italiano di Tecnologia, Rovereto 38068, Italy. [5]Center for Multimodal Imaging and Genetics, Department of Radiology, University of California, San Diego, La Jolla, CA 92093, USA. [6]Department of Chemistry and Biochemistry, University of California, San Diego, La Jolla, CA 92093, USA. ✉e-mail: kuaikuaiduan@gmail.com; ecourchesne1949@gmail.com

cortical organoids show overgrowth that is associated with poor language and social abilities and significant growth alterations in cortical language, social, and sensory regions in the same child's MRI scan data[22]. Non-cortical brain regions, however, show inconsistent brain alteration patterns in ASD; for example, both volume increases and reductions have been reported in the amygdala[23–25], corpus callosum[26–29], and cerebellum[30–33]. The inconsistent results may be due to cohort (e.g., subject characteristics such as age and clinical phenotype), MRI scanner, preprocessing pipeline, and analytical methodology differences[25,34].

Most studies focused on global measures or single regions (e.g., amygdala, cerebellum, and corpus callosum) and single morphometries (e.g., volume, surface area, cortical thickness) of interest that may be relevant to ASD. However, in one recent study of the correlation between gene co-expression modules and cortical thickness and surface area of different cortical regions in ASD and typical toddlers, we found evidence of atypical anterior-posterior and dorsal-ventral genomic cortical patterning in ASD toddlers with cortical enlargement and poor language outcome[35]. Genes in relevant modules enrich neural progenitor cells; symmetrical and asymmetrical cell division that can alter surface area and cortical thickness; and excitatory neurons, oligodendrocyte precursors, endothelial cells, and microglia that may affect thickness[35]. In a review of ASD genetic[36], postmortem, and animal and cell models, we concluded ASD is a disorder involving progressive disruption of many different prenatal stages in cortical regional growth, including area, lamina, and cellular development. Moreover, ASD child and adult postmortem cortex show substantial regional differences in the number and function of differentially expressed (DE) genes[37] and that opens the possibility of regional differences in volume, area, and thickness across ASD cortex. Thus, while the ASD cortex may be enlarged overall, genetic dysregulation of multiple processes across prenatal stages and across different cortical regions at older ASD ages, could lead to diverse types of deviances in volume, surface area, and thickness across cortical regions. In the cortex, surface area (SA) and cortical thickness are dissociable features[38]; examining potential alterations in both features in the same sample may point to distinct biological origins of cortical gray matter changes. With the exception of one small sample ($n = 15$) study[39], no study of brain alterations in fully diagnosed and confirmed ASD participants at the age of first early detection has yet examined regional differences across the brain and examined volume, cortical thickness, and SA in a comprehensive manner.

Brain size alterations have been widely reported to underlie language and social deficits and facial recognition impairment in ASD. For example, volumes in frontal and temporal regions were related to repetitive behavior and social and communication deficits in ASD, as revealed in an unbiased voxel-based morphometry study[30] or a source-based morphometry (a multivariate approach) study[40]. Moreover, Dziobek and colleagues identified that increased cortical thickness in the fusiform gyrus was associated with more severe face processing impairments in middle-aged adults with autism[41]. These studies used a cross-sectional design and examined an older sample among whom compensatory neural alterations may have resulted from behavioral challenges. Fusiform dysfunction in response to social stimuli has been found in numerous imaging studies[42–46], suggesting possible anatomic differences as well.

Increased temporal cortical thickness in ASD with lower intellectual ability and more severe symptoms have been found in MRI analyses of a wide-age range of ASD participants in a multi-site study[46]. Temporal cortex dysfunction in response to social language has been replicably shown in four independent cohorts of ASD toddlers at the time of first clinical detection and diagnosis[47–51], also suggesting underlying structural abnormality. The priori prediction of temporal cortex structural aberrance is also strongly bolstered by the fact that the temporal cortex is a major hot spot for ASD gene dysregulation,

harboring 2,733 dysregulated genes, which is 6X to 20X more than all other ASD cortices except visual and parietal BA7[37].

Beyond the effort to identify early-age anatomic growth abnormalities in ASD, is the major clinical translation challenge of discovering anatomic predictors of the heterogeneous developmental outcomes in ASD wherein some ASD toddlers get better, while others get worse with age[35,49,50]. Despite the importance of this, it is yet not clear whether structural alterations of cortical and subcortical regional size at first clinical detection in toddlers can contribute to discriminating different prognosis trajectories. Here we test outcome predictors using combined MRI and clinical variables, analogous to our prior study of combined fMRI and clinical variables[49].

Another major question in the field of ASD brain development is whether variation in deviant growth in specific anatomic regions relates to ASD symptom severity, language ability, and cognition at early-age diagnostic intake. Given the previous structural and functional ASD studies described above, variation in temporal, fusiform, and frontal anatomic measures are likely to have ASD clinical correlations. In previous studies, variation in temporal cortex dysfunction is related to variation in language and social measures[49,51]. Specifically, there is a predictable brain-behavior relationship: patients with more reduced social fMRI activation in the temporal cortex, have more severe ASD social symptoms and more reduced language abilities[51]. Therefore, just as ASD toddlers with the most dysfunctional temporal cortex, have the lowest language and social outcomes[51], we hypothesize a corresponding relationship will be found between MRI structural measures of temporal cortex and ASD symptoms and language ability in toddlers with ASD. Replicated across scores of studies, the temporal cortex is a hot spot for social information processing in typical adults[52–60] as well as typical toddlers[51,61–63]. Its reliable activation in typical sleeping toddlers and adults but reduced activation in ASD toddlers—even during sleep when attention, task demands, arousal, coorperation are not confounders—strongly supports the hypothesis that this region will display anatomical variation at early ages related to variation in symptom and language dysfunction in ASD toddlers.

In this work, we first examine complete and replicable regional early brain alterations in a large sample of $N = 166$ ASD and $N = 109$ typically developing (TD) toddlers. Specifically, we comprehensively and systematically investigate differential regional brain volume and cortical SA and thickness measurements in ASD compared to TD toddlers, while controlling for global brain size. We then examine the replicability of discovered regional differences in an independent toddler cohort (38 ASD, 37 TD) using the same preprocessing pipeline and the same statistical methods. Next, we investigate whether including regional size measures found to be altered at early intake age (mean = 2.4 years) improves a model's ability to predict language outcome at 6-month longitudinal follow-up testing beyond intake clinical and behavioral measures. Lastly, we test the hypothesis that variations in temporal, fusiform, and specific frontal measures correlate with variations in language and social symptom severity in ASD toddlers. Finding such a within-ASD correlation could open future clinical translational research using quantitative MRI to index ASD clinical features at the time of initial detection and diagnosis.

Here, we show in contrast to TD toddlers, toddlers with ASD have larger or thicker temporal and fusiform regions; smaller or thinner inferior frontal lobe and midline structures; larger callosal subregion volume; and smaller cerebellum even after factoring out brain sizes. The majority of the identified brain alterations are replicated. Moreover, these brain alterations enhance accuracy for predicting language outcome at 6-month follow-up beyond intake clinical and demographic measures. Brain alterations in temporal, fusiform, and inferior frontal are associated with autism symptom severity and cognitive impairments at early intake ages. To sum, brain regions involved in language, social, and face processing are altered in toddlers with ASD. Measures of these regional anatomical alterations improve the

prediction accuracy for future language ability and index autism symptom severity and cognition deficits.

## Results

### Brain-size adjusted ASD vs. TD brain structure difference in the main sample

Early brain overgrowth in ASD is one of the widely reported findings on ASD brain structural development[12–16,18,19]. Hence, we started by examining the ASD vs. TD difference of global brain measures (i.e., the estimated total intracranial volume (eTIV), total cortical SA, and mean cortical thickness) in our samples using linear mixed effect models (LMEMs, see details in Methods) while adjusting effects from age, sex (fixed effect) and longitudinal scans (random intercept and slope for each subject). In the main sample ($N = 275$, see Table 1 for their demographic and clinical characteristics at the time of initial scanning), no significant ASD vs. TD difference was observed for eTIV ($p = 0.96$), total cortical volume ($p = 0.07$), total cortical SA ($p = 0.49$), or mean cortical thickness ($p = 0.47$). Note that all brain measures tested in the manuscript met the assumptions of LMEM models and the residual formed normal distributions.

Our previous study with a smaller sample size demonstrated that ASD toddlers with poor early-age language outcomes (ASD Low, Mullen expressive and receptive language T score <40 at 3–4 years of age) had significantly enlarged cortical volume and cortical SA compared to TD[35]. Thus, we examined whether this ASD Low vs. TD difference was also present in the current study. We found out that the ASD Low subtype also presented significantly greater total cortical volume compared to TD toddlers ($p = 2.56 \times 10^{-3}$, Cohen's d (referred to as d hereafter, 95% confidence interval (95% CI)) = 0.39 (0.15, 0.63), beta = 8.62).

Using LMEMs (see details in Methods) while adjusting effects from age, sex, brain global measurements (fixed effects) and longitudinal scans (random intercept and slope for each subject), we found four cortical regions had significant volume differences between ASD and TD toddlers after FDR at $p < 0.05$ correction (Fig. 1 upper left): ASD toddlers had significantly increased gray matter volume (GMV) in left hemisphere (LH) fusiform ($p = 2.23 \times 10^{-4}$, d (95% CI) = 0.42 (0.21, 0.63), beta = 0.50); LH middle temporal ($p = 9.09 \times 10^{-4}$, d (95% CI) = 0.38 (0.17, 0.59), beta = 0.46); and right hemisphere (RH) middle temporal ($p = 2.87 \times 10^{-5}$, d (95% CI) = 0.49 (0.27, 0.70), beta = 0.62) regions compared to TD toddlers. Also, ASD toddlers had significant GMV reduction in RH caudal anterior cingulate compared to TD ($p = 7.07 \times 10^{-4}$, d (95% CI) = −0.39 (−0.60, −0.18), beta = −0.18).

Moreover, four cortical regions showed a significant thickness difference between ASD vs. TD toddlers (Fig. 1 upper right). Compared to TD, ASD toddlers had significantly thicker cortex in LH superior

temporal ($p = 1.05 \times 10^{-4}$, d (95% CI) = 0.45 (0.23, 0.66), beta = $5.24 \times 10^{-3}$) and RH banks of the superior temporal sulcus (bank STS) ($p = 1.49 \times 10^{-3}$, d (95% CI) = 0.37 (0.16, 0.58), beta = $7.06 \times 10^{-3}$) regions. Compared to TD, ASD had significantly thinner cortex in LH

**Table 1 | Demographic information and intake clinical test scores for ASD and TD toddlers in main sample**

| Characteristics | ASD (166 toddlers) | TD (109 toddlers) | p value (ASD vs. TD) |
|---|---|---|---|
| **Demographics at MRI and clinical visit** | | | |
| Sex (M/F) | 137/29 | 65/44 | $2.60 \times 10^{-5}$ [a] |
| Age at clinical visit (years) | 2.40 (0.69) | 1.91 (0.77) | $2.71 \times 10^{-7}$ [b] |
| Age at MRI scan (years) | 2.50 (0.69) | 2.05 (0.75) | $6.49 \times 10^{-7}$ [b] |
| **ADOS (module T, 1 or 2) score** | **ASD ($N = 166$)** | **TD ($N = 106$)** | |
| ADOS SA | 13.77 (4.43) | 1.96 (2.22) | $1.30 \times 10^{-69}$ [c] |
| ADOS RRB | 3.87 (1.93) | 0.28 (0.64) | $2.95 \times 10^{-46}$ [c] |
| ADOS Total | 17.64 (5.55) | 2.25 (2.45) | $6.04 \times 10^{-74}$ [c] |
| **Mullen score** | **ASD ($N = 161$)** | **TD ($N = 98–101$)** | |
| Ratio fine motor (ratio FM)[d] | 86.05 (17.40) | 111.93 (14.04), $N = 99$ | $4.40 \times 10^{-17}$ [c] |
| Ratio visual reception (ratio VR)[d] | 87.25 (19.58) | 116.33 (16.82), $N = 98$ | $7.59 \times 10^{-19}$ [c] |
| Ratio expressive language (ratio EL)[d] | 63.89 (22.32) | 104.46 (19.42), $N = 100$ | $1.63 \times 10^{-30}$ [c] |
| Ratio receptive language (ratio RL)[d] | 64.43 (24.40) | 110.93 (19.84), $N = 98$ | $1.73 \times 10^{-33}$ [c] |
| Early learning composite (ELC) | 73.27 (17.80) | 111.75 (17.36), $N = 101$ | $7.94 \times 10^{-35}$ [c] |
| **Vineland standard score** | **ASD ($N = 166$)** | **TD ($N = 107$)** | |
| Adaptive behavior composite | 80.44 (9.99) | 102.96 (10.07) | $4.73 \times 10^{-43}$ [c] |
| Daily living skills | 84.22 (11.20) | 103.26 (10.66) | $7.09 \times 10^{-29}$ [c] |
| Socialization | 81.05 (10.85) | 104.21 (9.64) | $2.36 \times 10^{-43}$ [c] |
| Motor skills | 90.61 (11.38) | 99.99 (8.97) | $2.12 \times 10^{-9}$ [c] |
| Communication | 77.45 (13.97) | 102.94 (11.08) | $5.20 \times 10^{-39}$ [c] |

All statistical tests were two-tailed. Values for age and all clinical test scores are presented as mean (SD). SD represents standard deviation. ADOS SA represents ADOS social affect and ADOS RRB presents ADOS restricted and repetitive behavior. ADOS, Mullen, and Vineland are evaluated at the same clinic visit.
[a]Pearson's chi-squared test.
[b]Welch's two-sample t-test.
[c]N-way ANOVA test including age and sex as covariates.
[d]Mullen subscale ratio score was computed by dividing the age equivalent score of that subscale by the toddler's chronological age.

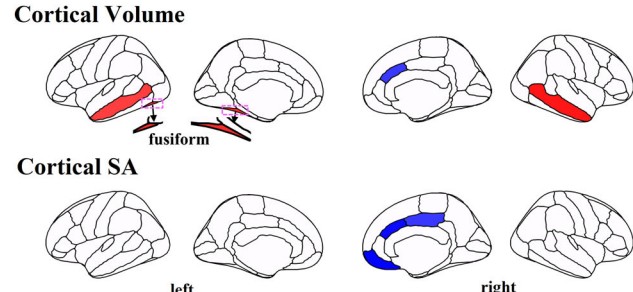

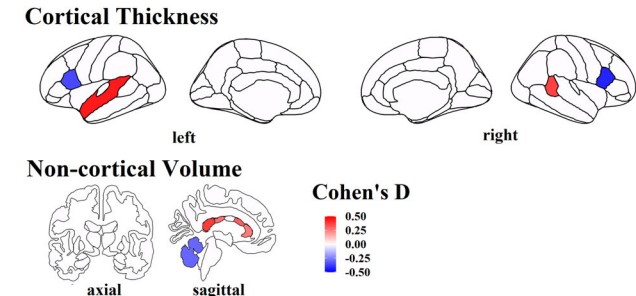

**Fig. 1 | Brain regions showing significant differences between ASD and TD toddlers in the main sample in terms of cortical volume, non-cortical volume, cortical thickness, and cortical SA.** Colors represent corresponding effect sizes (Cohen's D), where regions with hot colors showed significant increases in size among ASD compared to TD and regions with cold colors showed significant decreases in size among ASD compared to TD; the darker the color, the larger the difference between ASD and TD. Cortical regions showing significant volume difference (upper left) include LH fusiform, LH and RH middle temporal, and RH caudal anterior cingulate. Cortical regions showing significant thickness difference (upper right) include LH superior temporal, RH bank STS, LH, and RH pars opercularis. Cortical regions showing significant SA difference (lower left) include RH caudal anterior cingulate, RH medial orbitofrontal, and RH posterior cingulate. Non-cortical regions showing significant volume difference (lower right) include RH cerebellum, posterior CC, mid posterior CC, mid anterior CC, and anterior CC.

pars opercularis ($p = 1.68 \times 10^{-3}$, d (95% CI) = −0.36 (−0.57, −0.15), beta = −4.49 × 10⁻³) and RH pars opercularis ($p = 2.76 \times 10^{-4}$, d (95% CI) = −0.43 (−0.64, −0.22), beta = −5.77 × 10⁻³).

In terms of SA, ASD toddlers showed significantly reduced cortical SA compared to TD in RH caudal anterior cingulate ($p = 3.99 \times 10^{-5}$, d (95% CI) = −0.48 (−0.69, −0.27), beta = −0.45), RH medial orbitofrontal ($p = 2.46 \times 10^{-5}$, d (95% CI) = −0.50 (−0.71, −0.28), beta = −0.59) and RH posterior cingulate ($p = 1.01 \times 10^{-3}$, d (95% CI) = −0.39 (−0.60, −0.18), beta = −0.39) regions (Fig. 1 lower left).

Outside of the cerebral cortex, ASD toddlers presented significantly increased volume compared to TD in posterior CC ($p = 1.23 \times 10^{-3}$, d (95% CI) = 0.38 (0.17, 0.59), beta = 0.04), mid posterior CC ($p = 1.58 \times 10^{-2}$, d (95% CI) = 0.28 (0.07, 0.49), beta = 0.01), mid anterior CC ($p = 7.32 \times 10^{-3}$, d (95% CI) = 0.31 (0.10, 0.52), beta = 0.02), and anterior CC ($p = 2.73 \times 10^{-2}$, d (95% CI) = 0.26 (0.05, 0.47), beta = 0.02). ASD toddlers also showed decreased GMV in the right cerebellum compared to TD ($p = 9.39 \times 10^{-3}$, d (95% CI) = −0.30 (−0.51, −0.09), beta = −1.13) (Fig. 1 lower right). No significant ASD vs. TD difference was observed for subcortical regional GMV ($p > 0.05$ for all subcortical regions).

Stratifying by sex, the identified ASD vs. TD volume differences were still present in both males and females in the cerebellum, posterior CC, LH fusiform, LH and RH middle temporal, and RH caudal anterior cingulate ($p < 0.05$, see details in **Sex-stratified ASD vs. TD brain structure differences in main sample** in Supplementary Material). Moreover, compared to TD, toddlers with ASD had smaller surface area in RH caudal anterior cingulate and smaller cortical thickness in RH pars opercularis in both females and males ($p < 0.05$, Supplementary Table 1). The remaining subregional surface area and cortical thickness differences were present in males ($p < 0.05$) but not in females (Supplementary Table 1).

When only including the initial MRI scan for each of the 275 toddlers, we observed that most of the identified brain measures still displayed ASD vs. TD differences, although some were with weaker effect sizes compared to those from inclusion of repeated scans, but all except mid posterior CC and anterior CC had $p$ values less than 0.05 (see details in **ASD vs. TD brain structure differences for initial MRI scans** in Supplementary Material).

Age square did not affect the identified ASD vs. TD differences (See details in **Age square effect on ASD vs. TD brain structure differences in main sample** in Supplementary Material). Violin plots of brain regions showing significant ASD vs. TD differences are displayed in Supplementary Fig. 1. Results of brain regions showing nominal significant ASD vs. TD differences ($p < 0.05$) are listed in Supplementary Data 1.

## Brain-size adjusted ASD vs. TD brain structure differences in the replication sample

The diagnostic, sex, and age characteristics of the replication sample are listed in Table 2 (psychometric assessments in Table 1 were not available for the replication sample). The replication sample was collected between 1–5 years old, and their scan age was significantly older than that of the main sample ($p = 1.22 \times 10^{-4}$). The male/female ratio was comparable between ASD and TD toddlers in the replication sample ($p = 0.55$), while the ASD group has significantly more males compared to TD group in the main sample.

In the replication sample, ASD toddlers had significantly bigger brains (eTIV, $p = 0.02$, d (95% CI) = 0.45 (0.15, 0.76), beta = 12.55), greater total cortical volume ($p = 2.22 \times 10^{-3}$, d (95% CI) = 0.58 (0.27, 0.89), beta = 16.76) and larger mean cortical thickness ($p = 1.20 \times 10^{-4}$, d (95% CI) = 0.77 (0.45, 1.08), beta = 7.66 × 10⁻³) compared to TD. No significant ASD vs. TD difference was observed for total cortical SA ($p = 0.08$). Three out of four cortical regions that had significant GMV differences between ASD and TD toddlers in the main sample were replicated (Fig. 2 left): ASD toddlers had significantly increased GMV in LH fusiform ($p = 9.53 \times 10^{-3}$, d (95% CI) = 0.52 (0.21, 0.83), beta = 0.54), LH middle temporal ($p = 1.55 \times 10^{-7}$, d (95% CI) = 0.97 (0.65, 1.29), beta = 1.19) and RH middle temporal ($p = 1.87 \times 10^{-4}$, d (95% CI) = 0.70 (0.39, 1.02), beta = 0.95) regions compared to TD toddlers. Among regions showing significant thickness differences between ASD and TD toddlers in the main sample, three were replicated (Fig. 2 middle): Compared to TD, ASD toddlers had significantly thicker cortex in LH superior temporal ($p = 3.02 \times 10^{-7}$, d (95% CI) = 1.02 (0.70, 1.34), beta = 1.15 × 10⁻²) and RH bank STS ($p = 0.05$, d (95% CI) = 0.37 (0.06, 0.67), beta = 6.68 × 10⁻³) regions, and significantly thinner cortex in the LH pars opercularis ($p = 5.33 \times 10^{-3}$, d (95% CI) = −0.51 (−0.82, −0.20), beta = −6.10 × 10⁻³) region. Moreover, compared to TD, toddlers with ASD showed significantly increased volume in mid anterior CC ($p = 1.22 \times 10^{-3}$, d (95% CI) = 0.58 (0.27, 0.89), beta = 0.04), mid posterior CC ($p = 6.12 \times 10^{-3}$, d (95% CI) = 0.55 (0.24, 0.86), beta = 0.02), and anterior CC ($p = 5.11 \times 10^{-3}$, d (95% CI) = 0.55 (0.24, 0.86), beta = 0.05), ASD toddlers also had decreased volume in right cerebellum cortex ($p = 1.55 \times 10^{-2}$, d (95% CI) = −0.41 (−0.71, −0.10), beta = −1.82) (Fig. 2 right). None of the three cortical regions showing significant SA differences were replicated ($p > 0.05$). Violin plots of brain regions that were replicated for ASD vs. TD differences are presented in Supplementary Fig. 2.

## Early regional brain sizes improve ASD language outcome prediction

As previously employed for prognostic analyses[35,49,50], we stratified language outcome of ASD toddlers as ASD Low/Average or ASD Low based on Mullen expressive language (EL) and receptive language (RL) T scores at the outcome visit. An ASD toddler was grouped as ASD Low if both Mullen EL and RL T scores were below −1 SD of the T score norm of 50 (i.e., T < 40). An ASD toddler was classified as ASD Low/Average if

**Table 2 | Demographic of replication samples**

| Demographics | ASD (38 toddlers) | TD (37 toddlers) | p value (ASD vs. TD) |
|---|---|---|---|
| Sex (M/F) | 29/9 | 26/11 | 0.55[a] |
| Age (years) | 3.11 (0.80) | 2.33 (0.89) | 1.51 × 10⁻⁴ [b] |

Values for age are presented as mean (SD). SD represents standard deviation.
[a]Pearson's chi-squared test.
[b]Welch's two-sample t-test.

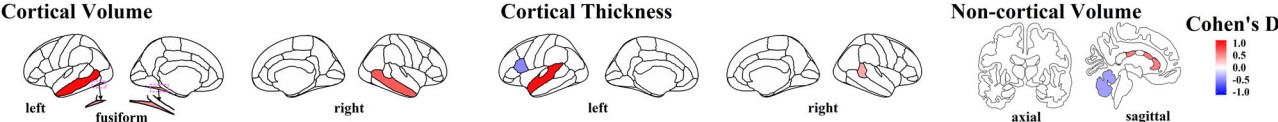

**Fig. 2 | Brain regions replicated for ASD vs. TD differences in cortical volume, non-cortical volume, and cortical thickness.** Colors represent corresponding effect sizes (Cohen's D), where regions with hot colors showed significant increases in size among ASD compared to TD and regions with cold colors showed significant decreases in size among ASD compared to TD; the darker the color, the larger the difference between ASD and TD. Cortical volume differences (left) are replicated in LH fusiform, LH, and RH middle temporal. Cortical thickness differences (middle) are replicated in LH superior temporal, LH pars opercularis, and RH bank STS. Non-cortical volume differences (right) are replicated in the RH cerebellum, mid posterior CC, mid anterior CC, and anterior CC.

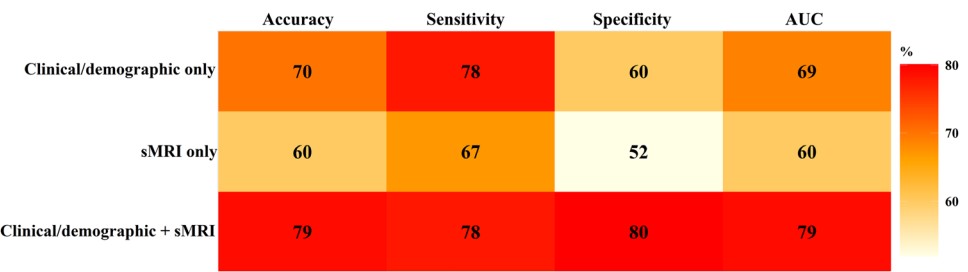

**Fig. 3 | Accuracy, sensitivity (for detecting ASD low), specificity (for detecting ASD low/average), and AUC values of clinical/demographic-only, sMRI-only, clinical/demographic + sMRI models for predicting ASD low/average vs. low** **language outcome.** Features used in each model were collected at the initial visit (the earliest clinical visit, mean age = 2.35 years). Language outcome was evaluated at a mean age of 2.85 years.

the toddler had either Mullen EL or RL T scores equal to or greater than −1 SD of the normative T score of 50 (i.e., T ≥ 40). Out of 166 ASD toddlers, 157 had a Mullen evaluation at the outcome visit and were stratified into two outcome groups: ASD Low/Average (N = 69; 59 males, 10 females; age = 2.82 ± 0.37 years) and ASD Low (N = 88; 71 males, 17 females; age = 2.88 ± 0.43 years). These 157 ASD toddlers were further used for language outcome prediction analysis. Their Mullen EL and RL T scores at outcome visit are displayed in Supplementary Fig. 3, where ASD Low/Average toddlers showed no significant difference for language outcome compared to TD (p = 0.61 for Mullen ELT, p = 0.35 for Mullen RLT, see details in **Language outcome differences between ASD Low/Average and TD** in Supplementary Material).

We employed a support vector machine (SVM) with ridge regularization to classify future language outcomes (ASD Low/Average or ASD Low). We tested and evaluated three different models: clinical/demographic-only, structural MRI (sMRI) only, and clinical/demographic + sMRI models (see details in Methods). We used sMRI, clinical/demographic features collected at the initial visit (mean age = 2.35 years) to classify language outcomes half year later. Each model was trained and cross-validated with the training samples (80% samples, N = 124 (54 ASD Low/Average toddlers, 70 ASD Low toddlers), female/male = 21/103, baseline age: 2.36 ± 0.67 years, outcome age: 2.87 ± 0.39 years) using five-fold cross-validation, and its performance was evaluated with an untouched hold-out testing set (20% samples, N = 33 (15 ASD Low/Average toddlers, 18 ASD Low toddlers), female/male = 6/27, baseline age: 2.34 ± 0.67 years, outcome age: 2.81 ± 0.46 years). Figure 3 plots the performance of clinical/demographic-only, sMRI-only, and clinical/demographic + sMRI models for classifying ASD low/average vs. low language outcome. Sensitivity and specificity reflect the accuracy for correctly detecting ASD Low and ASD low/average, respectively. Combining intake clinical/demographic and sMRI features yielded the highest accuracy (79%) and area under the receiver operating characteristic curve (AUC = 79%) compared to that from a single modality (sMRI-only model: accuracy = 60%, AUC = 60%; clinical/demographic-only model: accuracy = 70%, AUC = 69%). Both the clinical/demographic + sMRI model and the clinical/demographic-only model had a sensitivity (ASD low detection) of 78%. Including sMRI features to the model, the specificity (ASD low/average detection) improved from 60% (the clinical/demographic-only model) to 80% (the clinical/demographic + sMRI model); AUC improved from 69 to 79%; Accuracy improved from 70 to 79%. Supplementary Fig. 4 displays the contribution (weight) of each baseline clinical/demographic and sMRI feature to predicting the language outcome of ASD toddlers.

Supplementary Fig. 5 plots the histogram of AUC, accuracy, sensitivity, and specificity values from 100 iterations of five-fold cross-validation for the Clinic + sMRI model. The frequently observed AUC, accuracy, and sensitivity values were around 80%, 79%, and 94%,

respectively, which were close to or larger than the reported values (AUC = 79%, accuracy = 79%, sensitivity = 78%). Some iterations even achieved better performance than the reported one.

## Associations between behavior and growth-aberrant temporal, fusiform, and inferior frontal regions in ASD toddlers

First comprehensively confirming anatomic abnormality of temporal, fusiform, and inferior frontal regions and then showing the significant prognostic value of combining anatomic measures, symptom and cognitive variables, we next determined how these specific cortical regions are correlated with early-age ASD symptoms (ADOS) and cognition (Mullen ELC, EL, RL, and VR), a major specific question in the field of ASD brain development, as stated in the Introduction. To test this specific early-age question of whether growth in these regions index the most important clinical phenotypic characteristics, we used intake MRI measures and behavioral assessments closely matched to the intake MRI visit.

Four cortical volume measures were significantly related to ADOS symptom severity or Mullen subscale scores after FDR correction (Fig. 4 and Supplementary Fig. 6). In ASD toddlers (Fig. 4), larger (i.e., more aberrant) GMV in LH fusiform was significantly associated with higher ADOS total (i.e., more severe symptoms, r (95% CI) = 0.17 (0.02, 0.32), p = 2.62 × 10⁻²), and lower Mullen ELC (r (95% CI) = −0.25 (−0.39, −0.10), p = 1.51 × 10⁻³), lower Mullen ratio RL (r (95% CI) = −0.28 (−0.42, −0.14), p = 2.76 × 10⁻⁴) and lower Mullen ratio VR scores (r (95% CI) = −0.26 (−0.40, −0.11), p = 9.58 × 10⁻⁴) (i.e., poorer performance on Mullen subscales). Similarly, larger (i.e., more aberrant) GMVs in LH and RH middle temporal were significantly associated with higher ADOS SA (LH: r (95% CI) = 0.24 (0.09, 0.38), p = 1.74 × 10⁻³; RH: r (95% CI) = 0.24 (0.09, 0.38), p = 2.30 × 10⁻³) and higher ADOS total (LH: r (95% CI) = 0.24 (0.09, 0.38), p = 1.79 × 10⁻³; RH: r (95% CI) = 0.24 (0.09, 0.38), p = 2.06 × 10⁻³) scores, and lower Mullen ELC (LH: r (95% CI) = −0.19 (−0.33, −0.03), p = 1.87 × 10⁻²; RH: r (95% CI) = −0.20 (−0.35, −0.05), p = 9.94 × 10⁻³), lower Mullen ratio RL (LH: r (95% CI) = −0.19 (−0.33, −0.03), p = 1.94 × 10⁻²; RH: r (95% CI) = −0.20 (−0.34, −0.05), p = 1.15 × 10⁻²) and lower Mullen ratio VR (LH: r (95% CI) = −0.19 (−0.33, −0.03), p = 1.86 × 10⁻²; RH: r (95% CI) = −0.18 (−0.33, −0.02), p = 2.35 × 10⁻²). This LH middle temporal-Mullen ratio VR association was opposite to that seen in TD (see details in **Associations between brain structures and behavior in TD toddlers** in Supplementary Material). Paradoxically, larger (i.e., less aberrant) SA in RH caudal anterior cingulate was significantly related to lower Mullen ratio RL (r (95% CI) = −0.22 (−0.36, −0.06), p = 6.05 × 10⁻³).

## Associations between behavior and growth-aberrant non-cortical brain structures in ASD toddlers

Five non-cortical regional volume measures that had significant differential volumes in ASD vs. TD analyses in the main sample, were

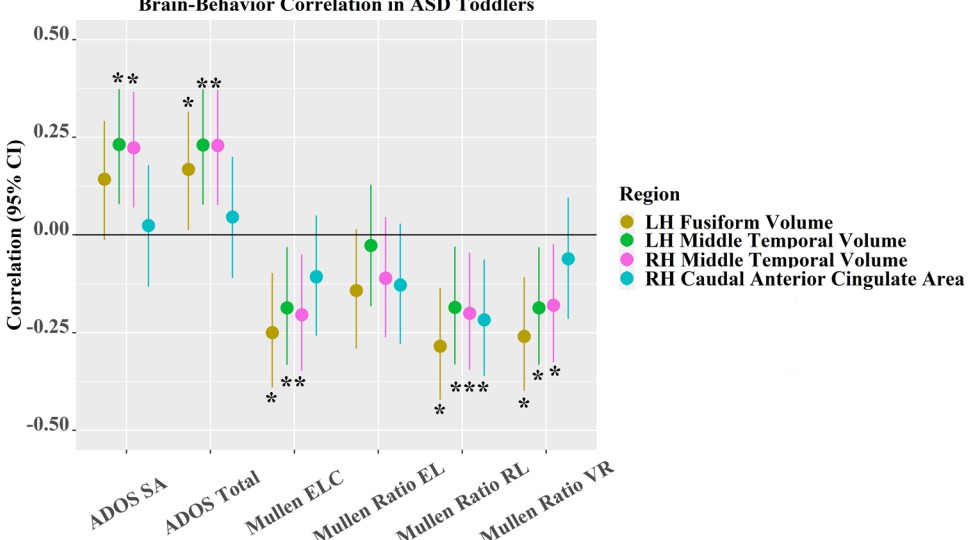

**Fig. 4 | Associations between behavior and ASD discriminating cortical regions in ASD toddlers and its 95% CI.** Note that * indicates the correlation is significant; colors of medium dark shades of yellow, green, cyan and a medium light shade of magenta denote LH fusiform volume, LH middle temporal volume, RH caudal anterior cingulate SA, and RH middle temporal volume, respectively. $N = 166$ independent toddlers with ASD were tested for associations with ADOS or Vineland subscales. $N = 161$ independent toddlers with ASD were examined for associations with Mullen subscales. The dot represents the true correlation value, and the error bar represents its 95% confidence interval.

tested for correlations with ADOS, Mullen (ELC, RL, EL), and vineland (adaptive behavior composite and daily living skills) scores as a posteriori analysis. While these regions have often been found in ASD to differ from controls, there is no literature strongly suggesting the index variation in the degree of symptom severity or cognition impairment in heterogeneous autism groups at early ages. Again, we only used intake MRI measures and behavioral assessments closely matched to the intake MRI visit.

Three out of five non-cortical volume measures were significantly related to Mullen or Vineland subscale scores in ASD toddlers after FDR correction (Fig. 5 and Supplementary Fig. 6). Larger GMV in posterior CC ($r$ (95% CI) = −0.25 (−0.39, −0.10), $p = 1.38 \times 10^{-3}$) and mid anterior CC ($r$ (95% CI) = −0.20 (−0.34, −0.04), $p = 1.23 \times 10^{-2}$) was significantly associated with lower Mullen ratio RL. This mid anterior CC-Mullen ratio RL association was opposite to what was observed in TD toddlers (see details in **Associations between brain structures and behavior in TD toddlers** in Supplementary Material). Larger GMV in the right cerebellum cortex was significantly associated with higher Vineland adaptive behavior composite ($r$ (95% CI) = 0.23 (0.08, 0.37), $p = 3.45 \times 10^{-3}$) and Vineland daily living skills ($r$ (95% CI) = 0.32 (0.17, 0.45), $p = 4.93 \times 10^{-5}$).

## Discussion

In this study, we surveyed the volume, thickness, and surface area of all regions across the brain to observe which size measures were reproducibly altered in ASD toddlers compared to TD toddlers. Identified brain regions are mainly involved in receptive and expressive language, social and face processing (bank STS, middle temporal, superior temporal, medial orbitofrontal, caudal anterior cingulate, posterior cingulate, and pars opercularis)[42–45,59,60,64–68]. Additional regions included those involved in motor, behavioral, cognitive, language control, and interhemispheric communication (cerebellum and corpus callosum)[69–75]. Morphometrically, we observed alterations in regional volume, thickness, and surface area relative to global measures. Thus, by first factoring out brain size, differentially increased or decreased growth in different anatomic measures in ASD-relevant language, social, face processing, and behavior regulation regions were isolated and highlighted. Cortically, lateral temporal regions

tended to be larger or thicker in ASD than TD; frontal lobe and midline structures tended to be smaller or thinner in ASD. Outside the cortex, larger callosal subregion volume and smaller cerebellum were observed. The majority of the identified GMV and cortical thickness alterations were replicated in an independent cohort. Importantly, larger (i.e., more aberrant) GMV in LH and RH middle temporal and LH fusiform were related to more severe ADOS social symptoms and/or poorer Mullen cognitive performance (ELC, ratio RL, and ratio VR) in ASD toddlers. Also of clinical relevance, the identified brain features measured at early intake ages, when included in a predictive model along with clinical and demographic features, markedly improved the accuracy for classifying low/average vs. low language outcome for toddlers with ASD at 6-month longitudinal follow-up clinical testing. The results from 100 iterations of five-fold cross-validation for the clinic + sMRI model indicate that the reported classification performance is unlikely driven by a single "lucky" splitting of the data.

The identified regional alterations were largely consistent with previous findings. Studies have found that young children[2,18,19,21], adolescents, and adults[76] with ASD show GMV enlargement in the temporal lobe, especially in the superior and middle temporal and fusiform gyri[77]. Increased cortical thickness in left hemisphere superior temporal cortex (LH STC) also appears to be a very strong and replicable finding in the literature, as evident in other large-scale studies in primarily adolescents and adults[46,78]. The current results showcase that increased LH STC thickness is present even earlier in ASD in toddlerhood and with larger effect sizes than studies in older ASD individuals. This developmentally ubiquitous increase in cortical thickness of LH STC may yield insight into early developmental processes that contribute to cortical thickness (e.g., proliferation of excitatory neuronal cell types in different cortical layers). Furthermore, normative brain charts indicate that cortical thickness tends to peak in early childhood followed by a slow decline over the lifespan[79], so these ASD toddler results combined with others in older ASD samples would indicate that increased early developmental cortical thickening combined with attenuated cortical thinning of LH STC may be a robust and key neural feature of ASD neurodevelopment. Given the observations of early developmental functional abnormalities in LH STC for language[49–51], these converging results may indicate that atypical neural

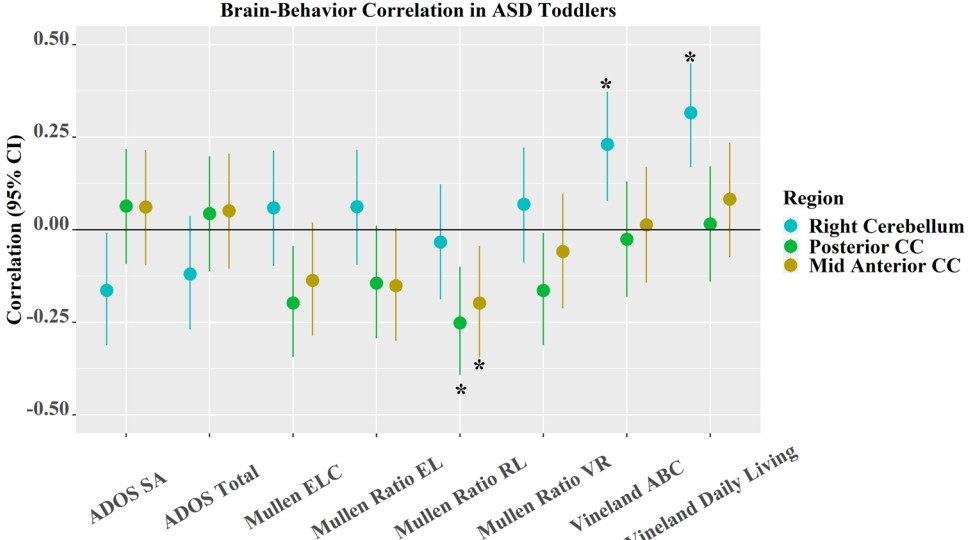

**Fig. 5 | Association between behavior and ASD discriminating non-cortical brain regions in ASD toddlers and its 95% CI.** Note that * indicates the correlation is significant; colors of cyan, green, and medium dark shades of yellow denote the right cerebellum, posterior CC, and mid anterior CC, respectively. $N = 166$ independent toddlers with ASD were tested for associations with ADOS or Vineland subscales. $N = 161$ independent toddlers with ASD were examined for associations with Mullen subscales. The dot represents the true correlation value, and the error bar represents its 95% confidence interval.

development and underlying genomic mechanisms affecting LH STC[35,50] may be the cause of reduced functional activation of this region by social affective language in ASD 1 to 3 year olds and atypical social-communication development.

GMV reduction in the cerebellum has been well-documented for individuals with ASD spanning from childhood to late adulthood[30–32]. Postmortem studies also reveal that individuals with ASD have decreased number[80] and reduced size[81] of Purkinje cells in the cerebellar hemisphere and vermis. The identified volume increase in CC in ASD toddlers aligns with the finding that infants with ASD have significantly increased SA and thickness in CC starting at 6 months of age, and the increase is particularly robust in the anterior CC at both 6 and 12 months[27]. Other studies[26–29] suggest that CC in individuals with ASD likely undergoes overgrowth at early ages[27], followed by abnormally slow or arrested growth, and later shows a reduction in adulthood[28,29]. Our results of SA reduction in the orbitofrontal cortex and posterior cingulate are consistent with a recent study led by Ecker[82]. Moreover, the identified alterations in thickness align with the finding by ref. 83 that individuals with ASD have reduced thickness in the left pars opercularis (the opercular part of the inferior frontal gyrus) during childhood and adolescence as well as in the right pars opercularis during adulthood.

While ASD toddlers had increased cortical volume, by first factoring out the overall size, we revealed a differential abnormality in cortical patterning in multiple ASD-relevant language, social, face processing, and behavior regulation regions. Abnormality was manifest in a complex map of differentially increased or decreased GM volume, surface area and thickness and highlights the presence of dysregulated regional cortical growth. These early-age regional alterations of cortical attributes may be the result of progressive dysregulation in multiple neural and molecular processes and stages, consistent with prenatal multi-process, multi-stage models of ASD[36,84].

One mechanism that could be involved in these effects is the overactivity of a prenatal multi-pathway gene network, a gene dysregulation that is present in ASD-derived prenatal progenitors and neurons and related to early-age ASD social symptom severity[85]. This gene network, the differentially expressed (DE)-ASD Network, is composed of DE genes in ASD toddlers, and includes PI3K-AKT, RAS-ERK, Wnt, and Insulin receptor signaling pathways and upstream regulatory ASD risk genes. These signaling pathways normatively have a strong impact on prenatal brain patterning and development because they regulate proliferation, neurogenesis, differentiation, migration, neurite outgrowth, and synaptogenesis[84,86–90]. The overactivity of gene expression in this DE-ASD Network is present in ASD vs. typical toddler progenitors and neurons and is greater in ASD toddlers who have more severe social symptoms[85]. Based on BrainSpan data (http://www.brainspan.org), this network normatively expresses during the first and second trimesters in multiple cortical areas during cortical patterning and progenitor cell division and neurogenesis[85]. Future studies should focus on the relationships between gene dysregulation in this DE-ASD Network in living ASD toddlers, brain cortical organoid models, and the toddlers' neural and clinical phenotype to test this potential mechanism.

We found that toddlers with ASD who had more aberrant brain measures also showed poorer cognitive performance and more severe ADOS-based social symptoms. The identified brain-behavior associations largely align with previous findings. Recently, Grecucci and colleagues[40] reported that larger GMV in an autism-specific structural network (including fusiform and middle temporal gyri) was related to higher ADOS symptom subscales (social affect and restricted and repetitive behavior) and total scores. Rojas et. al[30] also reported that GMV in the temporal region was positively associated with social and communication total symptom score. A study led by Dziobek reported that increased cortical thickness in the fusiform gyrus was related to impairments in face processing in adults with ASD[41], consistent with our result that fusiform GMV was negatively related to the Mullen ratio VR score. Brain-behavior associations that were not consistent between ASD and TD groups suggest that the neurobiology of social communication in ASD may differ from that observed in TD.

In the present work, the identified brain regions were highly valuable for characterizing prognosis. The sMRI-clinical/demographic combined model achieved the highest accuracy for classifying ASD low/average vs. ASD low, which parallels our functional imaging finding that a multimodal fMRI-clinical model outperformed single modality models[49]. Integrating multiple modalities can take full advantage of both modality-unique and complementary information from other modalities that is key for parsing ASD heterogeneity. Notably, although the sMRI model had the highest accuracy (sensitivity) for detecting ASD low, the accuracy for ASD low/average was low. There were two possible reasons: (1) our samples included more ASD Low than ASD

low/average toddlers, resulting in better detection of ASD low characteristics than that of ASD low/average; and (2) the features input to SVM were more pronounced in ASD low than ASD low/average in general (See Supplementary Tables 8–10), although a few showed reversed patterns.

The findings presented in this study should be considered in context with its strengths and limitations. Using brain regions showing significant ASD vs. TD differences as input for SVM reduced the likelihood of overfitting of the model. However, we may have missed other features that might also be important for discriminating ASD low/average from ASD low. Future research should include a full exploration of all FreeSurfer features and train a nested cross-validated machine learning model to select top features to improve the accuracy and reproducibility for detecting ASD low/average. Another future direction is to train and test more sophisticated regression models by including all FreeSurfer features and exploring different feature reduction techniques to quantitively predict future language outcomes. Another limitation is that while a majority of the identified brain alterations were replicated, further replication with larger samples is still necessary, especially for regions showing SA differences. Although we have carefully adjusted the age effect in the model, given our TD toddlers were 5 months younger than ASD toddlers, there may be remaining development effects confounding group differences; future replication samples with even closer age matching can test this possibility. Moreover, the mismatches of sample size, age, and sex between main and replication samples may have weakened the power to replicate some identified brain regional differences even though we adjusted for effects from age, sex, and global brain measurements in both main sample and replication sample. One potential limitation is that given that participants received the original version of the ADOS (ADOS-2 was not available at the time of evaluation), calibrated symptom severity scores for the Toddler Module were not available. Thus, ADOS total scores were used in analyses, with potential minor differences in overall total scores across modules.

In summary, ASD toddlers showed GM alterations in regions mainly involved in language, social, and face processing. Most identified GM alterations were replicated in an independent cohort. These early-age brain alterations may be the result of dysregulation in multiple neural processes and stages. Moreover, the identified GM alterations were predictive of ASD future language ability when combined with intake clinical measures, and they were presented as indices of social and language at the early age of first detection.

## Methods

This study was approved by the Institutional Review Board at the University of California, San Diego (UCSD). Written informed consent was obtained from parents or legal guardians for all toddlers included in this study. Parents or legal guardians were compensated for their participation.

### Main sample

All toddlers were recruited through community referrals or a general population-based screening method called Get SET Early[91], previously known as the 1-Year Well-Baby Check-Up Approach[92,93], allowing detection of ASD at early ages (e.g., -12 months). Toddlers were tracked from an intake assessment (1–3 years of age) and followed roughly every 12 months until 3 to 4 years of age (outcome visit). All toddlers participated in a series of clinical and behavioral assessments at each visit, including ADOS (Module T, 1, or 2) for ASD symptom evaluation[94–96], the Mullen Scales of Early Learning[97] for evaluating early cognition, and the Vineland Adaptive Behavior Scales[98] for assessing a child's functional skills in four different developmental domains.

All assessments were performed by licensed psychologists with PhD degrees and occurred at the UCSD Autism Center of Excellence.

Diagnosis at the most recent clinical visit was used in this study. Diagnosis of ASD is determined by highly experienced and licensed psychologists using diagnostic criteria in DSM IV[99] or V[1] in combination with the gold-standard ADOS evaluation[100]. TD toddlers showed no history of any developmental delay. Since some toddlers with ASD were scored at the floor of the standardized scores on Mullen subscales, we computed a ratio score for each subscale by dividing the age equivalent score by the toddler's chronological age[101–104]. We used these ratio scores to evaluate their associations with brain morphometry.

Clinical and behavioral scores and sMRI scans were collected from 343 toddlers (198 ASD and 145 TD). Around 30% of toddlers had follow-up sMRI scans collected, contributing to 447 scans in total. Among 343 toddlers, 68 had poor sMRI scans or scans with bad segmentation quality (see details later) and were excluded from the study, yielding data from 275 toddlers (166 ASD, 109 TD; 202 male, 73 female; $2.21 \pm 0.76$ years). ASD and TD groups showed no significant difference for maternal education ($p = 0.77$), sibling status ($p = 0.84$), number of siblings ($p = 0.50$), and median household income ($p = 0.11$). No significant female vs. male differences were observed for ADOS/Vineland and most Mullen (except Mullen ELC) subscale scores (see details in **Sex-stratified clinical test scores in main sample** section in Supplementary Methods). Out of 275 toddlers, 187 had only an intake sMRI scan collected, 88 had one or more follow-up sMRI scans collected at intervals ranging from 0.5 to 27 (mean ± standard deviation: $13.03 \pm 3.35$) months after the initial/previous scan, contributing to 372 scans in total (see details in **Distribution and time points of MRI scans in main sample** in Supplementary Methods). To boost statistical power, all 372 scans passed quality control were used to examine ASD vs. TD differences.

### MRI data acquisition and preprocessing

Imaging data were collected on a 1.5 T General Electric MRI scanner during natural sleep at night; no sedation was used. Structural MRI data were collected with a T1-weighted IR-FSPGR (inversion recovery fast-spoiled prepared gradient recalled) sagittal protocol with TE (echo time) = 2.8 ms, TR (repetition time) = 6.5 ms, flip angle = 12°, bandwidth = 31.25 kHz, field of view = 24 cm, and slice thickness = 1.2 mm. All sMRI scans were parcellated using FreeSurfer 5.3 (http://surfer.nmr. mgh.harvard.edu/)[105] based on the Desikan-Killiany atlas[106] to provide global and regional brain morphometric measures, including total brain volume, total surface area, mean cortical thickness, cortical subregional volume/SA/thickness, and subcortical volumes. FreeSurfer aligns each toddler's brain to an average brain derived from cortical folding patterns through nonlinear surface-based registration[107]. This tool has been validated for studies of children[108] and has shown great success in large pediatric studies[35,109,110]. Quality evaluation was further performed on the raw and segmented sMRI scans by two independent raters with a rating scale ranging from 0 to 3 (0=best, 1=great, 2=usable, 3=unusable). The inter-rater reliability for quality rating was estimated as 0.64 (CI: [0.48, 0.75]) using two-way random-effects ANOVA model[111,112]. Out of 447 sMRI scans, 75 were rated as unusable and were excluded from the study, yielding 372 scans. ASD and TD showed no difference in terms of the proportion of scans being identified as poor sMRI or bad segmentation quality (see details in **Quality rating of MRI scan and segmentation** in Supplementary Methods).

### Replication sample

Seventy-six toddlers (38 ASD and 38 TD) recruited in our previous study[18] were used as a replication sample. Toddlers were recruited through clinical referrals or advertisements and were diagnosed by the same licensed psychologist with the abovementioned criteria. sMRI scans were collected at the same site with a 1.5 T Siemens Symphony system during the toddler's natural sleep at night. A total of 170 sMRI

scans were collected at intake and follow-up visits. All replication sMRI scans were preprocessed with FreeSurfer 5.3 using the same pipeline and the same Linux platform as used for the main samples. Similarly, sMRI scans with excessive motion or bad segmentation quality were excluded, yielding 167 scans from 75 unique toddlers (38 ASD, 37 TD; 55 male, 20 female) for testing the replicabilities of ASD vs. TD differences identified from the main sample. The detailed participant recruitment, diagnosis evaluation, and scan collection information can be found in ref. 18.

## Brain structure difference between ASD and TD toddlers

For both the main and replication samples, ASD vs. TD differences in global and regional brain size were examined using the same (two-tailed) LMEMs ("fitlmematrix.m" in Matlab 2020b) as described later. Assume we have M unique participants and N (initial and follow-up) MRI scans, brain global measurement (eTIV, total cortical SA, and mean cortical thickness) differences between ASD and TD were tested using the LMEM:

$$\text{Brain global measure}_{im} = \beta_0 + \beta_1 \times diagnosis_i + \beta_2 \times \text{scan age}_i + \beta_3 \times sex_i + b_{0m} + b_{1m} \times scanage_{im} + \varepsilon_{im}.$$

where $i = 1, 2, ..., N$, and $m = 1, 2, ..., M$. $\beta_j$ ($j = 0, 1, 2, 3$) were the fixed-effects coefficients, and $b_{0m}$ and $b_{1m}$ were random-effects coefficients (i.e., the intercept $b_{0m}$ and coefficient of scan age $b_{1m}$ varied by subject). The random effects and observation error had the prior distributions: $b_{0m} \sim N(0, \sigma_0)$, $b_{1m} \sim N(0, \sigma_1)$, $\varepsilon_{im} \sim N(0, \sigma^2)$. Here each global brain measure was treated as the dependent variable, and fixed-effect predictors included diagnosis, age at scan, and sex. Scan age was treated as a random effect to take longitudinal scans into account. The diagnosis was coded as a dummy variable (ASD = 1, TD = 0). Thus, for each brain region tested, the beta value of diagnosis can be interpreted as how much larger/smaller (unit: cm for thickness, cm² for SA, cm³ for volume) ASD toddlers' brains are compared to TDs' brains. ASD vs. TD differences in cortical and subcortical volume, cortical regional surface area and thickness were tested using the LMEM as below:

$$\text{Regional volume/SA/thickness}_{im} = \beta_0 + \beta_1 \times diagnosis_i + \beta_2 \times \text{scan age}_i + \beta_3 \times sex_i + \beta_4 \times \text{brain global measure}_i + b_{0m} + b_{1m} \times scanage_{im} + \varepsilon_{im}.$$

where $i = 1, 2, ..., N$, and $m = 1, 2, ..., M$. Volume/SA/thickness of each brain region was treated as the dependent variable. Scan age was treated as a random effect and $b_{0m}$ and $b_{1m}$ were random-effects coefficients (each subject had a random intercept and random slope for scan age). $\beta_j$ ($j = 0, 1, 2, 3, 4$) were the coefficients for fixed-effects predictors (diagnosis, age at scan, sex, and brain global measure). Brain global measures included eTIV for testing subcortical and cortical regional volume, total cortical SA for testing regional SA, and mean cortical thickness for testing regional thickness measures. To identify cortical regions with significant volume/SA/thickness differences between ASD and TD in the main sample, a false discovery rate (FDR) at $p < 0.05$ was applied to correct for 204 comparisons (68 LH and RH cortical regions, three measures (volume/SA/thickness) for each cortical region). FDR at $p < 0.05$ was also applied to correct for comparisons of subcortical regions, cerebellum (LH and RH), and corpus callosum (CC) regions separately. The identified ASD vs. TD differences were considered as replicated if the corresponding $p$ values were less than 0.05 in the replication sample. Studies have suggested that brain may undergo nonlinear processes in early ages[79,113], we further tested quadratic age effect on ASD vs. TD brain structure differences by including age² as a covariate (fixed effect) in the above LMEM model.

Given that recent work has reported sex differences in brain structure in ASD[114–118], we further performed sex-stratified ASD vs. TD

difference tests for significant cortical and subcortical volume, cortical regional surface area, and thickness using the abovementioned LMEM while excluding sex from the fixed-effect predictors. Moreover, we examined whether the identified significant regional differences hold when only the initial MRI scan was included for each participant to rule out any effects from repeated (initial/follow-up) measures using the linear regression model:

$$\text{Regional volume/SA/thickness} = \beta_0 + \beta_1 \times diagnosis + \beta_2 \times \text{scan age} + \beta_3 \times sex + \beta_4 \times \text{brain global measure} + \varepsilon.$$

## Predicting language outcome for ASD toddlers

We employed SVM with ridge regularization to predict future language outcome. SVM with ridge can select features of importance to achieve a stable classification result. We tested and evaluated three different models: clinical/demographic-only, sMRI-only, and clinical/demographic + sMRI models. The clinical/demographic-only model used behavioral (ADOS, Mullen, and Vineland) and demographic (sex, age at intake, and gap between intake and outcome visit) variables at intake visit (i.e., only baseline measures were used). The sMRI-only model leveraged age and sex-adjusted intake FreeSurfer measures (age and sex effects were estimated using TD data[119]) within regions that showed significant ASD vs. TD differences. The clinical/demographic + sMRI model used all intake features (baseline) included in clinical/demographic-only and sMRI-only models. Each variable/feature was scaled to be between 0 and 1 prior to SVM for all models. Each model was trained and cross-validated with the training samples (80% samples) using fivefold cross-validation using "fitclinear.m" and "kfoldLoss.m" in Matlab 2020b. Its performance was evaluated with an untouched hold-out testing set (20% samples) and the predicted language outcome was computed using "predict.m" in Matlab 2020b. Accuracy, sensitivity, specificity, and AUC were computed for the untouched hold-out testing sample to reflect the performances of classification models. To reflect the dispersion of the classification performance of the clinical/demographic + sMRI model, we run 100 iterations of fivefold cross-validation and evaluated the classification performance on the untouched hold-out testing sample for each iteration.

## Brain-behavior association analyses

To test correlations between ASD discriminating temporal, fusiform, and frontal anatomic measures and ASD intake symptom severity and language measure (ADOS and Mullen ELC, RL, EL, and VR), we used the linear regression model ("regstats.m" in Matlab 2020b):

$$\text{Behavioral measure} = \beta_0 + \beta_1 \times volume/SA/thickness \text{ of a brain region} + \beta_2 \times age + \beta_3 \times sex + \varepsilon.$$

where each behavioral measure is treated as the response variable, and age, sex, and volume/SA/thickness of a brain region were predictors. In the brain-behavior association analyses, only the initial MRI measures and closely matched behavioral measures are used to maximize the sample size. FDR at $p < 0.05$ was applied to correct for comparisons from cortical regions showing volume, SA, and thickness differences separately. Associations between ASD differential non-cortical brain measures and ADOS symptoms, Mullen and Vineland subscales were also tested using the same regression model, and the results were corrected for multiple comparisons from non-cortical regions using FDR at $p < 0.05$.

## Reporting summary

Further information on research design is available in the Nature Portfolio Reporting Summary linked to this article.

## Data availability

Structural MRI and clinical data for main samples are available from the National Institute of Mental Health Data Archive via https://nda.nih.gov/edit_collection.html?id=9. The replication dataset uses existing data from refs. 18,79,120 and is available via the public repository: https://github.com/Luckykathy6/ASDLanguagePredict; https://zenodo.org/records/11200676[121]. The main dataset has been published in refs. 35,79,120.

## Code availability

Analysis scripts are available in the public repository: https://github.com/Luckykathy6/ASDLanguagePredict; https://zenodo.org/records/11200676[121].

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

## Acknowledgements

This work was supported by NIDCD grant R01DC016385 awarded to E.C. and K.P.; NIMH grants R01MH118879 and R01MH104446 awarded to K.P; and funding from the European Research Council (ERC) under the European Union's Horizon 2020 research and innovation program under grant agreement No 755816 to M.V.L. We would like to thank the parents and children in San Diego who participated in our research.

## Author contributions

K.D. and E.C. conceptualized the study. K.D. performed data analysis and wrote the manuscript. E.C., L.E., M.V.L., and K.P. helped interpret the results and revise the manuscript. D.J.H.Jr. helped with sMRI data segmentation. M.D. and K.C. helped with quality control on raw and segmented sMRI data. E.C., L.E., K.C., C.C.B., S.A., and S.N. contributed to data collection and data management. P.J. helped with the revision. E.C., K.P., L.E., and M.V.L. contributed to funding acquisition. All authors contributed to interpreting the results and discussion.

## Competing interests

The authors declare no competing interests.
