## [Peer Review File · Nature Communications]

Reviewers' Comments:

Reviewer #1:

Remarks to the Author:

Thank you for the opportunity to read and review the manuscript titled "Language, Social, and Face Regions Are Affected in Toddlers with Autism and Predictive of Language Outcome". The study reports on a large sample of 1-4 year old children with and without autism. Results include differences when comparing ASD to TD groups on cortical regions using measures of gray matter volume, surface area, and cortical thickness (results were small in effect size). Brain x behavior associations are also investigated as are prediction analyses. Strengths of the work include a large sample size. I also commend the authors for having a replication sample. However, the strengths and novelty of the work are diminished by missing information and choices in the statistical analysis that compromise the interpretation of the results.

1. There is missing information on the characterization of the sample. Is the age of the clinic visit that is reported in Table 1 the same as the age at "language outcome"? How many participants had the language outcome data? Lines 89-91 added confusion to this issue. The authors make a big deal about "prognostic" biomarkers but it is unclear how much earlier the scan really is to the language outcome measures. Further, according to Figure S6, MSEL RL and EL contributed the most weight to the model, so the "value add" of the scan is questionable.
2. Was the ADOS calibrated severity scores used? The table just states ADOS SA, ect. Since multiple modules were used the calibrated severity scores should have been used over the raw scores.
3. The age range is very large (12-50 months) in this study, which is a weakness. The authors include age as a covariate in their models, but there is plenty of research to indicate that brain development is not a linear process, so accounting for age linearly would be insufficient. Further, the ASD and TD groups were significantly different on age. This issue needs to be discussed in the limitations section.
4. What efforts were taken to evaluate motion in the scan data? Were there group differences in motion or in poor sMRI or bad segmentation quality? Can the authors please provide more information on how "poor sMRI" and "bad segmentation quality" was determined? How was reliability among the raters measured?
5. Some children in the study had repeated scans and the authors include all scans that passed quality control to "boost statistical power". While such an approach could increase power it also creates a lack of independence amongst observations within the analyses. Do the results hold if only one observation from each participant is included?
6. In the first paragraph of the results the authors look at total cortical volume in ASD participants with poor compared to TD toddlers. This analysis was not justified. Why was total cortical volume selected for this analysis? Why only compare TD to ASD with poor language? Were these a priori decisions? If so, how were the regions and comparisons selected?
7. The non-traditional approach to the brain x behavior analyses was not well justified. Why were the associations first tested within each group and then the interaction tested? A more parsimonious approach would be to test the interaction and simply have the within group tests of simple slopes nested into the main model. It's unclear exactly how many tests were done in this section and what the sample sizes looked like. Why was the ADOS SA tested but not the RRB score? Were only MSEL RL and VR tested? Why not EL, FM, GM?
8. I have some concerns about the prediction analyses. How were multiple observations handled for these analyses? I can only assume that each participant only contributed one scan to the prediction analysis, and if so, how was the scan selected? Were those decisions done a priori to analyses? What is the n for this analysis? Also, it is my understanding that the MSEL was used to create the ASD poor and ASD good groupings. But then the MSEL was also used to predict group membership for the analyses using clinical/demographic data. This "double-dipping" is problematic as it creates circular logic that results in over fitting. This is especially problematic as Figure S6 indicates that MSEL EL/RL contributed the most weight to the model.
9. More information is needed on the demographics and characteristics of the replication sample. Were the discovery and replication samples matched?
10. The reporting summary indicates that test statistics with confidence intervals, effect sizes, ect, are noted, but this is not inaccurate.

11. There are numerous places that need references: e.g., line 45, 95
12. There are numerous places where there are missing words: e.g., line 131, 168

Reviewer #2:

Remarks to the Author:

This study examined differences in gray matter structure in a large cohort of toddlers with and without ASD. Specifically, cortical/subcortical volume as well as cortical regional surface area and thickness were the main neural measurements of interest. Significant differences were observed for these measures between toddlers with and without ASD; some of these differences were replicated in a smaller independent cohort of toddlers with and without ASD. Interestingly, some of these brain alterations were related to ASD symptom severity and cognitive scores. Lastly, the authors conducted SVM using 3 models to predict language outcome (classified as a binary good vs poor language outcome): clinical/demographic only, sMRI only, and clinical/demographic + sMRI. The combined model outperformed the other two models with the highest accuracy and AUC metrics, though the models utilizing a single modality had higher specificity (clinical/demographic model) and sensitivity (sMRI). This was an interesting study and the results are compelling. In particular, the inclusion of a replication sample reinforces the robustness of the findings of altered gray matter volume. This manuscript stands to make a significant contribution to the field. Detailed comments are listed below.

- 1) Abstract: Explicitly state the ages/timepoints for the data (brain imaging and clinical/behavioral assessments) in the abstract.
- 2) Introduction: The authors clearly link the neural measures with possible underlying cellular/molecular processes in the Discussion. I would like to see some of this in the Introduction as well to set up the biological context for the neural measures that were chosen.
- 3) Introduction: In the last paragraph, it would be helpful to have a description of the ages at which the ADOS/Mullen/Vineland were collected.
- 4) Results: Include a demographic table for the replication sample as well.
- 5) The authors covary for sex in all analyses, but were there any differences in ADOS/Mullen/Vineland scores between males vs. females? I understand that the authors may not have the N to conduct analyses stratified by sex, but it would be interesting to see, particularly in light of recent work highlighting sex differences in brain function/structure in ASD. In Table 1, it would be helpful to see clinical/behavioral scores stratified by sex in addition to a description of the ages at which these measures were collected. Relatedly, would the prediction models look different for males vs females?
- 6) Only the gray matter volume analyses were conducted in the main and replication samples (i.e., the brain-behavior analyses and prediction analyses were only conducted in the main sample). Is because the clinical/behavioral data was not available for the replication sample? The rationale for this should be clearly stated.
- 7) Expand the discussion as to why certain regions but not others replicated. Is there something specific about the regions that did not replicate vs the regions that did?
- 8) It is interesting that the brain differences were predictive of receptive but not expressive language, but this was not discussed. Was this expected/unexpected? What could be some reasons as to why this was the case?
- 9) Methods: Were the ASD and TD groups matched on maternal education and sibling status (given implications for each of these factors on language learning/outcome)?
- 10) Methods: For each subject, it seems that multiple scans (all available scans) were used in the group-level analyses. Why was this decision made (vs using only one scan for more consistent ages at scan)? Why not look at longitudinal changes in brain structure if you have the data? What was the distribution of these datasets (how many subjects had multiple scans and at what timepoints)?

Minor:

- 1) Introduction, line 59: Is this a typo: "In this cortex" vs. "In this context" or "In the cortex"?
- 2) Introduction, lines 68-72: What ages were the subjects in the Dziobek study?
- 3) Figures 1 and 2: Include brain region names in the figure legends.

- 4) Discussion, lines 267-268: Specify inferior frontal gyrus for left/right pars opercularis.
- 5) Methods: Describe how Cohen's d was calculated.

Reviewer #3:

Remarks to the Author:

The paper by Duan and co authors, titled "Language, Social, and Face Regions Are Affected in Toddlers with Autism and Predictive of Language Outcome" and submitted to the journal Nature Communications, presents a study of the brain structural correlates of autism and how they relate to early language scores in toddlers. The strength of this work includes a large number of toddlers with repeated imaging and behavioral data. I have several comments listed below that I hope can improve the manuscript.

I have signed this review for transparency and am happy to discuss these comments if they are unclear. – Dustin Scheinost

Data from a 1.5T scanner is uncommon these days. Is there a reason for using a 1.5T scanner? Or is this older imaging data?

Given that some subjects had multiple scans, it may be useful to include a random slope in the LME model in addition to the random subject factor.

Relatedly, it could be helpful to include a table with how many subjects had a certain number of scans. For example, X subjects had y scans.

I am a bit unclear about the data used in the ASD Good vs ASD Poor analyses. The binarization of the data is based on Mullens at the 3 year old data collection. And the clinical factors to predict the Good / Poor distinction include Mullens scores from the previous two visits. Is this correct?

Why did the authors binarize based on Mullens outcomes? In theory, the authors could use a regression approach rather than classification to predict Mullen scores directly. That would provide more information in the subjects that are on the margin of the Good / Poor threshold.

Did the replication sample have follow up data to split them into Good / Poor to test the generalizability of the SVM classifier.

The authors correct for multiple comparisons over the number of regions analyzed. But they have three different imaging measures for each region (ie SA, volume, thickness). The authors should correct over these too.

In figure s5, the authors state that "ASD good toddlers had similar language outcome as TD toddlers". Did the authors do a direct comparison? If not, it would be interesting to see a ttest comparing scores for ASD good toddlers to TD toddlers.

For the classification analyses, it would be helpful for the authors to run multiple iterations of the random splitting into k folds and report the distribution of prediction performances. This will help the reader know the dispersion of the predictions and ensure that any performance is not due to a single "lucky" splitting of the data.

Dear Reviewers,

We appreciate your time and effort in helping us to improve the manuscript. We have revised our manuscript, “Language and Social Regions Are Affected in Toddlers with Autism and Predictive of Language Outcome”, according to your constructive and valuable comments. In the following, we provide detailed responses (in blue color) to each comment. Corresponding changes are highlighted in our revised manuscript.

Reviewer #1:

General comments:

Reviewer #1 (Remarks to the Author):

Thank you for the opportunity to read and review the manuscript titled “Language, Social, and Face Regions Are Affected in Toddlers with Autism and Predictive of Language Outcome”. The study reports on a large sample of 1-4 year old children with and without autism. Results include differences when comparing ASD to TD groups on cortical regions using measures of gray matter volume, surface area, and cortical thickness (results were small in effect size). Brain x behavior associations are also investigated as are prediction analyses. Strengths of the work include a large sample size. I also commend the authors for having a replication sample. However, the strengths and novelty of the work are diminished by missing information and choices in the statistical analysis that compromise the interpretation of the results.

Response: We thank the reviewer for the positive remarks and apologize for missing information and choices in the statistical analysis. We have carefully revised our manuscript according to reviewer’s comments. Please see the details below and in the revised manuscript (highlighted).

1. There is missing information on the characterization of the sample. Is the age of the clinic visit that is reported in Table 1 the same as the age at “language outcome”? How many participants had the language outcome data? Lines 89-91 added confusion to this issue. The authors make a big deal about “prognostic” biomarkers but it is unclear how much earlier the scan really is to the language outcome measures. Further, according to Figure S6, MSEL RL and EL contributed the most weight to the model, so the “value add” of the scan is questionable.

Response: Thank you for the comments and we are sorry for the confusion. The age of clinical visit in Table 1 is close to the age at the initial MRI scan (ASD clinic and MRI ages: 2.40 and 2.50 years; TD clinic and MRI ages: 1.91 and 2.05 years), and it is not the same as the age at language outcome (age: mean = 2.85 years). We have clarified this in the revised manuscript.

N=157 out of 166 ASD toddlers had language outcome data. Lines 89-91 were stating that the goal was to investigate whether initial regional brain measures showing ASD vs TD differences at *intake MRI ages* can improve the prediction accuracy of short-term language outcome at half year later. So, we use baseline MRI assessments, clinical and behavioral measurements collected at the initial early-age visit to predict those later short-term (about one half year) language outcome. We have clarified this in the revised manuscript (see highlighted). Yes, we agree that it is not surprising

that the baseline language ability (Mullen EL and RL) makes a strong contribution to predict future language outcome at 6 month follow up, but after including the MRI measurements, the overall accuracy substantially improved from 70% to 79% (AUC improved from 69% to 79%).

2. Was the ADOS calibrated severity scores used? The table just states ADOS SA, etc. Since multiple modules were used, the calibrated severity scores should have been used over the raw scores.

Response: Thanks for the comment. The ADOS Toddler Module does not have a calibrated severity score (CSS), only Modules 1-3 have. The CSS does not have a diagnostic cut-off per se. Only the ADOS total designates a cut-off for ASD. Moreover, the CSS only provides a level of ASD symptoms relative to other ASD patients. Thus, it does not provide a score for typical toddlers. These major limitations mean it is not diagnostically, quantitatively or statistically useful in our study.

3. The age range is very large (12-50 months) in this study, which is a weakness. The authors include age as a covariate in their models, but there is plenty of research to indicate that brain development is not a linear process, so accounting for age linearly would be insufficient. Further, the ASD and TD groups were significantly different on age. This issue needs to be discussed in the limitations section.

Response: 99.6% of our ASD and TD subjects (274 out of 275) had the initial scan collected at 1, 2 and 3 years of age. This is a very young and large sample of confirmed ASD. Only 3 of 37 ASD neuroimaging studies of toddlers are as large as our study. In our main sample, N=219 subjects were ages 1 and 2 years; N=55 were 3 years; and only N=1 subject was 4 years. Thus, we respectfully disagree that this is a “very large age range.” The age range is common in the field of neuroimaging of ASD toddlers as shown in Table 1, and our overwhelmingly young sample (N=219 ages 1 and 2, and N=55 age 3) is a great strength of the study. In fact, the age range of 1 to 3 or 4 years is relatively common across ASD neuroimaging studies that have been published in top journals (See Tables 1 and 2).

The age period of 1 to 3 years has high importance for understanding the development of ASD since it is during this period that (a) ASD is most commonly first detected and confirmed through longitudinal repeat diagnostic and psychometric testing, and (b) ASD toddlers display different clinical, behavioral, and brain developmental trajectories.

Additionally, of 37 ASD neuroimaging studies of toddlers, ours has largest and youngest total sample of confirmed ASD subjects and controls. The other large sample is aged 25 to 80 months. Please note that many neuroimaging studies with large young samples never confirmed their subjects had ASD and they therefore lack meaningful interpretation.

Table 2. Functional activation and Resting state fMRI Studies

fMRI Studies	ASD Sample	Contras t/ TD Sample	Age (months)
Courchesne Lab Studies	4 independent cohorts		
Lombardo et al., 2019	109	86	13 – 44
Lombardo et al., 2018	81	37	12 – 45
Lombardo et al., 2015	60	43	12 – 48
Xiao et al., 2022	41	30	14 – 55
Eyler, Pierce, & Courchesne, 2012	40	40	12 – 48
Redcay & Courchesne, 2008	12	23	13 – 47

Table 1. Structural MRI Studies

MRI Studies	ASD Sample	Contrast/ TD Sample	Age (months)
Courchesne Lab Studies			
Courchesne, Campbell, & Solso, 2011	259	327	1 – 50 years
Duan et al. current manuscript	204	146	12-50
Lombardo et al., 2021	76	47	12 – 50
Fingher et al., 2017	68	29	13 – 51
Courchesne et al., 2001	60	52	2 – 16 years
Akshoomoff et al., 2009	52	15	1 – 5 years
Schumann et al., 2010	41	44	12 – 48
Schumann, Barnes, Lord, & Courchesne, 2009	41	39	18 – 60
Bloss & Courchesne, 2007	36	27	1 – 5 years
Other Labs			
Reinhardt et al., 2020	200	110	25 - 80
Shiohama et al., 2021	181	0	0 - 36
Shen et al., 2018	159	77	24 - 48
Swanson et al., 2017	86	143	6 - 24
Gao, Sun, Niu, & Wang, 2021	74	208	24
MacDuffie et al., 2020	71	127	6 - 24
Hazlett et al., 2017	70	117	6 - 24
Cardenas-de-la-Parra et al., 2021	56	162	6 - 24
Shen et al., 2017	47	122	6 - 24
Xiao et al., 2017	46	39	18 - 37
Wolff et al., 2017	44	0	6 - 24
Conti et al., 2017	36	16	0 - 36
Fu et al., 2020	34	26	24 - 60
Lewis et al., 2017	31	76	6 - 12
Conti et al., 2016	20	0	20 - 36
Zhang et al., 2019	16	18	16 - 46

Other Labs			
Xu et al., 2019	93	79	2 – 5.5 years
Shen et al., 2016	43	29	Mean age 3.5 years
Marrus et al., 2018	~24	~50	12 – 24
Chen et al., 2021	24	23	17 – 45
McKinnon et al., 2019	20	39	12 – 24
Emerson et al., 2017	11	0	6 – 24
Liu et al., 2021	8	14	9 – 36

To account for possible nonlinear age effects, we included age² as a covariance in addition to age, sex, and global brain measures (eTIV, total surface area or mean cortical thickness). Tables 3-5 show the corresponding *p* and cohen's *d* values for identified regional volume, surface area and thickness measurements. We can see that most of the previously identified brain measures still show significant ASD vs. TD differences, and all have *p* values less than 0.05. We have included these results in the revised manuscript and supplemental material.

Table 3, ASD vs. TD difference of regional volume with including age² as a covariance.

Regions	p values	Cohen's d
lh_fusiform	2.20E-04	0.42
lh_middletemporal	1.79E-03	0.36
rh_caudalanteriorcingulate	9.21E-04	-0.38
rh_middletemporal	6.79E-05	0.46
Right-Cerebellum-Cortex	8.11E-03	-0.31
CC_Posterior	1.62E-03	0.37
CC_Mid_Posterior	2.02E-02	0.27
CC_Mid_Anterior	9.91E-03	0.30
CC_Anterior	3.16E-02	0.25

Table 4, ASD vs. TD difference of regional surface area with including age² as a covariance.

Regions	p values	Cohen's d
rh_caudalanteriorcingulate	5.18E-05	-0.47
rh_medialorbitofrontal	2.77E-05	-0.49
rh_posteriorcingulate	9.20E-04	-0.39

Table 5, ASD vs. TD difference of regional cortical thickness with including age² as a covariance.

Regions	p values	Cohen's d
lh_parsopercularis	1.60E-03	-0.36
lh_superiortemporal	3.21E-04	0.42
rh_bankssts	1.27E-03	0.38
rh_parsopercularis	2.61E-04	-0.43

In the discussion, we also added unmatched ages for TD and ASD as a limitation.

4. What efforts were taken to evaluate motion in the scan data? Were there group differences in motion or in poor sMRI or bad segmentation quality? Can the authors please provide more information on how “poor sMRI” and “bad segmentation quality” was determined? How was reliability among the raters measured?

Response: Thank you for the questions. Each scan was visually checked by two independent raters. Scans with obvious motion strips (see below for example, where arrows point to motion stripes) were flagged.

Among scans with poor sMRI or bad segmentation quality, 49.23% were from ASD toddlers and 50.77% from TD toddlers, indicating no group difference between ASD and TD. The inter-rater reliability for quality rating is estimated as 0.64 (CI: [0.48, 0.75]) using two-way random effects ANOVA model^{1,2}. Scans with highly visible motion stripes in T1w image or partial coverage of the brain were marked as poor quality. Bad segmentation quality refers to poor match between surfaces and aseg in multiple large regions of cortex (see below for example, where arrows demonstrate regions with poor matches).

5. Some children in the study had repeated scans and the authors include all scans that passed quality control to “boost statistical power”. While such an approach could increase power it also creates a lack of independence amongst observations within the analyses. Do the results hold if only one observation from each participant is included?

Response: Thanks for the comments. We included only the initial scan for each of the 275 toddlers and then tested ASD vs TD difference while controlling the effects from age, sex and global brain measures (eTIV, total surface area or mean cortical thickness). Tables 6-8 (see below) show the corresponding results, we can see that most of the previously identified brain measures still show ASD vs. TD differences, although some are with weaker effect sizes compared to that from inclusion of repeated scans, but all except mid posterior corpus callosum and anterior corpus callosum have p values less than 0.05. We have included these results in the revised manuscript and Supplement.

Combining ASD vs. TD difference results from replication samples, this indicates that including repeated scans while also carefully controlling the effects from repeated measures can help better identify brain regions related to ASD pathology that may be otherwise missed from smaller sample sizes.

Table 6, ASD vs. TD difference of initial regional volume.

Regions	p values	Cohen's d
lh fusiform	9.48E-03	0.35
lh middletemporal	4.80E-03	0.38
rh caudalanteriorcingulate	1.82E-02	-0.32
rh middletemporal	6.02E-04	0.46
Right-Cerebellum-Cortex	4.50E-03	-0.38
CC Posterior	1.55E-02	0.33

CC Mid Posterior	0.21	0.17
CC Mid Anterior	4.17E-02	0.27
CC Anterior	0.09	0.22

Table 7, ASD vs. TD difference of initial regional surface area.

Regions	p values	Cohen's d
rh_caudalanteriorcingulate	2.37E-03	-0.41
rh_medialorbitofrontal	4.48E-04	-0.48
rh_posteriorcingulate	7.37E-03	-0.36

Table 8, ASD vs. TD difference of initial regional cortical thickness.

Regions	p values	Cohen's d
lh_parsopercularis	3.31E-02	-0.29
lh_superiortemporal	5.20E-04	0.47
rh_bankssts	4.09E-02	0.28
rh_parsopercularis	1.68E-02	-0.32

6. In the first paragraph of the results the authors look at total cortical volume in ASD participants with poor language compared to TD toddlers. This analysis was not justified. Why was total cortical volume selected for this analysis? Why only compare TD to ASD with poor language? Were these a priori decisions? If so, how were the regions and comparisons selected?

Response: One of the most common statements written about ASD by countless researchers is that ASD is heterogeneous, and yet one of the most common mistakes is then putting all ASD subjects into one single group which then obliterates subtype differences and often other ASD effects of interest. Our approach has been different: Across the years we use methods to identify ASD clinical and neural subtypes, such as subtypes with poor language development or reduced social interest as indexed by eye tracking. These methods have revealed fMRI, MRI, genomic and eye tracking subtypes. In much smaller samples, we previously showed ASD toddlers with poor language had statistically larger cortices³. Therefore, since our goal was to reveal *differential regional* abnormality in cortical patterning in ASD *independent of overall size*, we first tested and verified this subtype effect was significant in our large sample and factored that out. This approach revealed a complex (and cortex-size-independent) map of differentially increased or decreased GM volume, surface area and thickness and highlighted the presence of dysregulated regional cortical growth.

7. The non-traditional approach to the brain x behavior analyses was not well justified. Why were the associations first tested within each group and then the interaction tested? A more parsimonious approach would be to test the interaction and simply have the within group tests of simple slopes nested into the main model. It's unclear exactly how many tests were done in this section and what the sample sizes looked like. Why was the ADOS SA tested but not the RRB score? Were only MSEL RL and VR tested? Why not EL, FM, GM?

Response: Thanks for the comments. In our previous ASD language-task functional MRI work⁴, heterogeneous brain-behavior association patterns were observed for TD and ASD subgroups, where ASD subgroups presented reversed brain function-behavior relationships compared to TD (Fig. 4A and 4B in ⁴). Given participants included in the current study were partially overlapped with those in ⁴, we anticipated different brain structure-behavior association patterns for ASD and TD groups. So, we tested them within ASD and TD group separately and then further tested the interaction effect. In this way, we fully captured regional brain structures underlying ASD

behavioral manifestations to enhance interpretation. We added the rationale to the revised manuscript (see highlighted).

We examined the associations between 3 ADOS, 5 Mullen and 5 Vineland subscales (as listed in Table 1 in the manuscript) and 11 cortical (4 volume, 3 SA and 4 thickness) measures and 5 non-cortical regional volume measures identified showing significant ASD vs. TD differences (detailed sample size for each subscale is added in the revised manuscript (see highlighted)). For this analysis, we only used intake MRI measurement and behavioral assessments closely matched to the intake MRI visit. No brain regions were significantly related to ADOS RRB, Mullen EL, Mullen FM or Vineland subscales. So, they were not reported. We have clarified this in the revised manuscript.

8. I have some concerns about the prediction analyses. How were multiple observations handled for these analyses? I can only assume that each participant only contributed one scan to the prediction analysis, and if so, how was the scan selected? Were those decisions done a priori to analyses? What is the n for this analysis? Also, it is my understanding that the MSEL was used to create the ASD poor and ASD good groupings. But then the MSEL was also used to predict group membership for the analyses using clinical/demographic data. This “double-dipping” is problematic as it creates circular logic that results in over fitting. This is especially problematic as Figure S6 indicates that MSEL EL/RL contributed the most weight to the model.

Response: Thanks for the comments. For the prediction analysis, we only used the *initial (baseline)* MRI scan, demographic and clinical measures closely matched with the initial MRI visit as features to predict language outcome. The follow-up MRI measures were not used in the prediction analysis. So, we used the baseline measurements collected at about 2.35 years of age to predict the language outcome at a 6-month follow-up clinical visit. Including baseline measurements as features to predict future ability has been used in many prognostic studies^{4, 5, 6, 7, 8}.

We are interested in predicting the *language outcome for ASD toddlers*, and the sample size for the prediction analysis was $N = 157$. We included the baseline MSEL scores to predict the future language outcome since we are interested in testing whether baseline language ability predicts future language outcome. It's not surprising that the baseline language ability has the largest contribution to predict future language outcome at around 6 month follow up.

We have used golden-standard strategies to train, validate and test the prediction model, where we randomly selected 80% samples to train and cross-validate the prediction model, and used remaining 20% samples as independent hold-out testing samples to test the performance of the trained model. The reported accuracies were obtained from independent hold-out testing samples. Overall, it is unlikely that the model is overfitted/overtrained.

9. More information is needed on the demographics and characteristics of the replication sample. Were the discovery and replication samples matched?

Response: We have age, sex and diagnosis information available for the replication sample. Please see detailed demographics in Table 9. The replication sample was collected for ages 1-5 years and so the MRI scan age is older than that of our main sample ($p = 1.22 \times 10^{-4}$). The male/female ratio was comparable between ASD and TD toddlers in replication sample ($p = 0.55$), while ASD group had significantly more males than girls compared to TD group in the main sample ($p = 2.60 \times 10^{-5}$). We have included the demographics of replication sample in the revised manuscript (highlighted). These several differences between main and replication samples –especially the small overall replication sample size– may underlie some differences in the number of significantly

different regions despite adjusting for effects of age, sex and global brain measurements in both main sample and replication sample. We have included these limitations in the discussion (highlighted).

Table 9, Demographic of replication samples.

Demographics	ASD (38 toddlers)	TD (37 toddlers)	p value (ASD vs. TD)
Sex (M/F)	29/9	26/11	0.55 ^a
Age (years)	3.11 (0.80)	2.33 (0.89)	1.51×10^{-4} ^b

^aPearson's chi-squared test.

^bWelch's two sample t test.

Note, values for age are presented as mean (SD). SD represents standard deviation.

10. *The reporting summary indicates that test statistics with confidence intervals, effect sizes, etc., are noted, but this is not inaccurate.*

Response: Thanks for the comments. For results of brain regional differences, we reported Cohen's *d* as the effect sizes. We have added 95% confidence interval for each Cohen's *d* value in the revised manuscript (see highlighted). For brain-behavioral association analyses, we reported correlation coefficient as the effect size and added 95% confidence interval in the bar plot (Fig. 3 and Fig. S3) and revised text (see highlighted). For outcome classification, since we used support vector machine, accuracy is reported, and no confidence interval can be obtained from support vector machine models.

11. *There are numerous places that need references: e.g., line 45, 95*

Response: Thanks for pointing that out. We have added references for those places.

12. *There are numerous places where there are missing words: e.g., line 131, 168*

Response: Thanks for the comments. We have added some words in those lines to make it clearer.

Reviewer #2 (Remarks to the Author):

This study examined differences in gray matter structure in a large cohort of toddlers with and without ASD. Specifically, cortical/subcortical volume as well as cortical regional surface area and thickness were the main neural measurements of interest. Significant differences were observed for these measures between toddlers with and without ASD; some of these differences were replicated in a smaller independent cohort of toddlers with and without ASD. Interestingly, some of these brain alterations were related to ASD symptom severity and cognitive scores. Lastly, the authors conducted SVM using 3 models to predict language outcome (classified as a binary good vs poor language outcome): clinical/demographic only, sMRI only, and clinical/demographic + sMRI. The combined model outperformed the other two models with the highest accuracy and AUC metrics, though the models utilizing a single modality had higher specificity (clinical/demographic model) and sensitivity (sMRI). This was an interesting study and the results are compelling. In particular, the inclusion of a replication sample reinforces the robustness of the findings of altered gray matter volume. This manuscript stands to make a significant contribution to the field. Detailed comments are listed below.

Response: We thank the Reviewer for the positive remarks.

1) *Abstract: Explicitly state the ages/timepoints for the data (brain imaging and clinical/behavioral assessments) in the abstract.*

Response: Thanks for the good suggestion. We have stated the ages and time points for the brain imaging and behavioral assessments in the abstract (see highlighted).

2) *Introduction: The authors clearly link the neural measures with possible underlying cellular/molecular processes in the Discussion. I would like to see some of this in the Introduction as well to set up the biological context for the neural measures that were chosen.*

Response: Thank you for this good suggestion. We have revised the Introduction and Discussion accordingly (see highlighted).

3) *Introduction: In the last paragraph, it would be helpful to have a description of the ages at which the ADOS/Mullen/Vineland were collected.*

Response: The ADOS/Mullen/Vineland were evaluated at the same clinical visit, and we have added the age that ADOS/Mullen/Vineland was collected in the last paragraph in Introduction (see highlighted).

4) *Results: Include a demographic table for the replication sample as well.*

Response: Thanks for the comment. Unfortunately, we only have age, sex and diagnosis information available for the replication sample. We have summarized their demographics in Table 9 and included in the revised manuscript (see highlighted).

Table 9. Demographic of replication samples.

Demographics	ASD (38 toddlers)	TD (37 toddlers)	p value (ASD vs. TD)
Sex (M/F)	29/9	26/11	0.55 ^a
Age (years)	3.11 (0.80)	2.33 (0.89)	1.51 × 10 ⁻⁴ b

^aPearson's chi-squared test.

^bWelch's two sample t test.

Note, values for age are presented as mean (SD). SD represents standard deviation.

5) *The authors covary for sex in all analyses, but were there any differences in ADOS/Mullen/Vineland scores between males vs. females? I understand that the authors may not have the N to conduct analyses stratified by sex, but it would be interesting to see, particularly in light of recent work highlighting sex differences in brain function/structure in ASD. In Table 1, it would be helpful to see clinical/behavioral scores stratified by sex in addition to a description of the ages at which these measures were collected. Relatedly, would the prediction models look different for males vs females?*

Response: Thanks for the comments. Given that sex collinears with diagnosis, and males are relatively older than female toddlers in main sample (Table 10), we tested male vs. female differences for ADOS/Mullen/Vineland subscale scores using N-way ANOVA tests including diagnosis and age as covariates. We found that ADOS/Vineland and most Mullen (except Mullen ELC) subscale scores did not show significant female vs. male differences (see detailed male vs. female differences in Table 10)). Only Mullen ELC showed a male vs. female difference, where females have higher ELC scores than males ($p = 0.04$). We have included these results in the revised manuscript and Supplement.

Table 10. Intake clinical test scores for male and female toddlers in main sample.

Characteristics	male (202 toddlers)	female (73 toddlers)	p value (male vs. female)
Demographics at MRI and clinical visit			
Number of ASD/TD	137/65	29/44	2.60 × 10 ⁻⁵ a
Age at clinical visit (years)	2.25 (0.76)	2.07 (0.75)	0.09 ^b
Age at MRI scan (years)	2.36 (0.76)	2.23 (0.71)	0.20 ^b
ADOS (module T, 1 or 2) score			
	male (N =202)	female (N = 72)	
ADOS SA	9.87 (6.74)	7.17 (6.85)	0.24 ^c
ADOS RRB	2.73 (2.30)	1.74 (2.31)	0.99 ^c

ADOS Total	12.60 (8.60)	8.90 (8.87)	0.34 ^c
Mullen score	male (N = 192-194)	female (N = 68-70)	
Ratio fine motor (ratio FM)	93.18 (20.50), N = 193	104.05 (18.33), N = 69	0.13 ^c
Ratio visual reception (ratio VR)	94.60 (22.17), N = 192	108.24 (23.35), N = 69	0.07 ^c
Ratio expressive language (ratio EL)	75.08 (26.62), N = 193	91.92 (31.50), N = 70	0.14 ^c
Ratio receptive language (ratio RL)	77.51 (29.91), N = 193	94.92 (34.17), N = 68	0.28 ^c
Early learning composite (ELC)	83.65 (23.80), N = 194	100.57 (26.57), N = 70	0.04 ^c
Vineland standard score	male (N = 202)	female (N = 73)	
Adaptive behaviour composite	87.37 (13.90)	94.77 (16.09)	0.52 ^c
Daily living skills	89.64 (13.70)	97.55 (14.67)	0.09 ^c
Socialization	88.49 (14.83)	94.96 (15.84)	0.72 ^c
Motor skills	93.71 (11.17)	96.10 (12.01)	0.68 ^c
Communication	85.39 (17.23)	93.29 (18.48)	0.61 ^c

^aPearson's chi-squared test.

^bWelch's two sample t test.

^cN-way ANOVA test including age and sex as covariates.

Note, values for age and all clinical test scores are presented as mean (SD). SD represents standard deviation. ADOS SA represents ADOS social affect, and ADOS RRB presents ADOS restricted and repetitive behavior. ADOS, Mullen and Vineland are evaluated at the same clinic visit.

We further tested ASD vs. TD differences in regional brain structure in males and females separately using linear mixed effect models while adjusting effects from age, brain global measurements (fixed effects) and longitudinal scans (random intercept and slope for each subject). The test results are listed in Table 11. The identified ASD vs. TD volume differences still show up in both males and females in cerebellum, posterior CC, LH fusiform, LH and RH middle temporal, and RH caudal anterior cingulate ($p < 0.05$). Moreover, compared to TD, ASD toddlers had smaller surface area in RH caudal anterior cingulate and smaller cortical thickness in RH pars opercularis in both female and males ($p < 0.05$). The remaining subregional surface area and cortical thickness differences were present in males ($p < 0.05$) but not females. We have included these results in the revised manuscript and supplement.

There are two possible interpretations: (1) these brain differences may potentially reflect sex differences or (2) given the small female sample size, we may be underpowered to detect the group differences in female.

Table 11. ASD vs TD brain structure differences stratified by sex.

ASD vs. TD difference	male (274 scans, 185 from ASD)		female (98 scans, 43 from ASD)	
	p	Cohen's d	p	Cohen's d
Non-cortical volume				
Right-Cerebellum-Cortex	0.04	-0.27	0.03	-0.45
CC_Posterior	2.84E-2	0.31	2.54E-2	0.52
CC_Mid_Anterior	0.12	0.22	2.47E-2	0.55
CC_Mid_Posterior	4.68E-2	0.28	0.32	0.24
CC_Anterior	0.29	0.15	3.15E-2	0.54
Cortical volume				
lh_fusiform	7.87E-3	0.36	6.14E-3	0.61
lh_middletemporal	4.96E-2	0.26	5.99E-4	0.90
rh_caudalanteriorcingulate	6.87E-3	-0.36	1.25E-2	-0.60
rh_middletemporal	4.63E-3	0.39	5.06E-4	0.76
Cortical surface area				
rh_caudalanteriorcingulate	1.73E-3	-0.42	2.06E-3	-0.75
rh_medialorbitofrontal	9.54E-5	-0.53	0.11	-0.38
rh_posteriorcingulate	6.58E-4	-0.48	0.81	0.05
Cortical thickness				
lh_superiortemporal	2.35E-5	0.58	0.51	0.15
lh_parsopercularis	9.90E-3	-0.35	0.08	-0.39
lh_caudalmiddlefrontal	2.78E-3	-0.40	0.62	-0.12
lh_pericalcarine	0.22	-0.17	0.10	-0.38
rh_parsopercularis	1.91E-3	-0.44	0.03	-0.48
rh_bankssts	2.98E-3	0.41	0.18	0.35

For 157 ASD toddlers used in language outcome prediction analysis, 130 are males and only 27 (17%) are female. So, prediction analysis in females is extremely underpowered.

6) *Only the gray matter volume analyses were conducted in the main and replication samples (i.e., the brain-behavior analyses and prediction analyses were only conducted in the main sample). Is this because the clinical/behavioral data was not available for the replication sample? The rationale for this should be clearly stated.*

Response: Yes, the behavioral data were not available for the replication sample. We have added the rationale in the revised manuscript.

7) *Expand the discussion as to why certain regions but not others replicated. Is there something specific about the regions that did not replicate vs the regions that did?*

Response: One explanation for lack of replication for some regions is small sample size in the replication sample. The main sample has 275 unique toddlers and 372 initial and follow-up MRI scans, while the replication sample only has 75 unique toddlers and 167 MRI scans. Another possible reason is the unmatched age and sex between main and replication sample. The replication sample was collected between 1-5 years old, but their scan age was significantly older than that of main sample ($p = 1.22 \times 10^{-4}$). The male/female ratio was comparable between ASD and TD toddlers in the replication sample ($p = 0.55$), while the ASD group had significantly more males than girls compared to TD group in the main sample ($p = 2.60 \times 10^{-5}$). Although we carefully and consistently adjusted age and sex effects in main and replication samples using the same LME models. There may be residual age and sex effects confounding the group differences. Replication samples with a larger sample size and matched age and sex are needed especially for regions not replicated. Nevertheless, regions that are replicated presented robust ASD vs. TD differences at different ages and different sex ratio settings in early development.

8) *It is interesting that the brain differences were predictive of receptive but not expressive language, but this was not discussed. Was this expected/unexpected? What could be some reasons as to why this was the case?*

Response: The ASD brain differences significantly related to social symptom and/or receptive language scores were cortical regions known to mediate social and/or language information processing. Interestingly, these fusiform and temporal regions had enlargement. These effects were predicted. We had also expected pars opercularis differences would be related to ASD expressive language, but we were surprised it was not.

9) *Methods: Were the ASD and TD groups matched on maternal education and sibling status (given implications for each of these factors on language learning/outcome)?*

Response: Yes. They are matched. ASD and TD groups showed no significant difference for maternal education ($p = 0.77$), sibling status ($p = 0.84$), number of siblings ($p = 0.50$), and median household income ($p = 0.11$). We have added this information in the revised manuscript.

10) *Methods: For each subject, it seems that multiple scans (all available scans) were used in the group-level analyses. Why was this decision made (vs using only one scan for more consistent ages at scan)? Why not look at longitudinal changes in brain structure if you have the data? What was the distribution of these datasets (how many subjects had multiple scans and at what timepoints)?*

Response: Thanks for the comments. We used all available initial and follow-up scans that passed quality control to boost statistical power, and we used linear mixed effect models to adjust random effects from repeated measures by including a random intercept and slope term for each subject in addition to adjusting fixed effects from age, sex and global brain measures (eTIV, total surface area or mean cortical thickness).

Additionally, we also only used the initial scan for each of the 275 toddlers and tested ASD vs TD difference with controlling the effects from age, sex and global brain measures (eTIV, total surface area or mean cortical thickness). Tables 6-8 (see below) show the corresponding results, we can see that most of the previously identified brain measures still show ASD vs. TD differences, although some are with weaker effect sizes compared to those from inclusion of follow-up scans, but all except mid posterior corpus callosum and anterior corpus callosum have *p* values less than 0.05. Combining ASD vs. TD difference results from replication samples, this indicates that including repeated scans while carefully controlling the effects from repeated measures can help better identify brain regions related to ASD pathology that may be otherwise missed from small sample sizes.

Table 6, ASD vs. TD difference of initial regional volume.

Regions	p values	Cohen's d
lh_fusiform	9.48E-03	0.35
lh_middletemporal	4.80E-03	0.38
rh_caudalanteriorcingulate	1.82E-02	-0.32
rh_middletemporal	6.02E-04	0.46
Right-Cerebellum-Cortex	4.50E-03	-0.38
CC_Posterior	1.55E-02	0.33
CC_Mid_Posterior	0.21	0.17
CC_Mid_Anterior	4.17E-02	0.27
CC_Anterior	0.09	0.22

Table 7, ASD vs. TD difference of initial regional surface area.

Regions	p values	Cohen's d
rh_caudalanteriorcingulate	2.37E-03	-0.41
rh_medialorbitofrontal	4.48E-04	-0.48
rh_posteriorcingulate	7.37E-03	-0.36

Table 8, ASD vs. TD difference of initial regional cortical thickness.

Regions	p values	Cohen's d
lh_parsopercularis	3.31E-02	-0.29
lh_superiortemporal	5.20E-04	0.47
rh_bankssts	4.09E-02	0.28
rh_parsopercularis	1.68E-02	-0.32

The longitudinal change is an interesting topic and is a future direction of the current research. The included 275 toddlers were initially scanned at a mean age of 27.98 months and longitudinally followed up at mean ages of 37.05, 44.04 and 55.10 months, respectively. The distribution of MRI scans was summarized in the Table 12. We have added it in the revised manuscript.

Table 12, The distribution of MRI scans in the main sample.

	1 scan	2 scans	3 scans	4 scans
Subject #	187	80	7	1

Minor:

1) Introduction, line 59: Is this a typo: “In this cortex” vs. “In this context” or “In the cortex”?

Response: Thanks for pointing out the typo. We have corrected it as ‘In the cortex’.

2) Introduction, lines 68-72: What ages were the subjects in the Dziobek study?

Response: In the Dziobek study, the mean age of ASD participant was 42 years, and the mean age of healthy controls was 45 years old. We have specified the subjects in the Dziobek study are middle-aged adults.

3) Figures 1 and 2: Include brain region names in the figure legends.

Response: We have now included the brain region names in the revised legend/caption for Figures 1 and 2.

4) Discussion, lines 267-268: Specify inferior frontal gyrus for left/right pars opercularis.

Response: Thanks for the comment. We have specified the inferior frontal gyrus for left/right pars opercularis. The revised sentence is: ‘Moreover, the identified alterations in thickness align with the finding by Zielinski et. al. ⁹ that individuals with ASD have reduced thickness in the left pars opercularis (the opercular part of inferior frontal gyrus) during childhood and adolescence as well as in the right pars opercularis during adulthood.’

5) Methods: Describe how Cohen’s d was calculated.

Response: The Cohen’s d and its confidence interval was computed using the ‘meanEffectSize.m’ function available in Matlab 2023a with data after subtracting fixed effects from confounding factors (e.g., age, sex and global brain measure) and random effects from repeated measures.

Reviewer #3 (Remarks to the Author):

The paper by Duan and co authors, titled “Language, Social, and Face Regions Are Affected in Toddlers with Autism and Predictive of Language Outcome” and submitted to the journal Nature Communications, presents a study of the brain structural correlates of autism and how they relate to early language scores in toddlers. The strength of this work includes a large number of toddlers with repeated imaging and behavioral data. I have several comments listed below that I hope can improve the manuscript.

I have signed this review for transparency and am happy to discuss these comments if they are unclear. – Dustin Scheinost

Response: We thank the reviewer for the positive remarks. Thanks for signing for transparent review and offering to discuss these comments.

Data from a 1.5T scanner is uncommon these days. Is there a reason for using a 1.5T scanner? Or is this older imaging data?

Response: Yes, this was part of a very large multimodality project completed about 8 years ago. At the time we started the project, UC San Diego only had a 1.5T research scanner and did not have a 3T scanner. Once the project started, we avoided the possible confounds that could result from using different scanners by finishing the project all on the same 1.5T scanner.

Given that some subjects had multiple scans, it may be useful to include a random slope in the LME model in addition to the random subject factor.

Response: Thanks for the suggestion. We have included a random slope in the LME model in addition to the random intercept for repeated scans. We have updated the results for the main sample and replication sample accordingly (highlighted in the revised manuscript). The previously identified significant ASD vs. TD differences still show up although with some small changes on p values and cohen's d values.

Relatedly, it could be helpful to include a table with how many subjects had a certain number of scans. For example, X subjects had y scans.

Response: Thanks for the suggestion. We now include a table (Table 12) to summarize the scan numbers.

Table 12, The distribution of MRI scans in the main sample.

	1 scan	2 scans	3 scans	4 scans
Subject #	187	80	7	1

The included 275 toddlers were initially scanned at a mean age of 27.98 months and longitudinally followed up at mean ages of 37.05 (scan 2), 44.04 (scan 3) and 55.10 (scan 4) months, respectively. We have included this information in the supplement.

I am a bit unclear about the data used in the ASD Good vs ASD Poor analyses. The binarization of the data is based on Mullen at the 3 year-old data collection. And the clinical factors to predict the Good / Poor distinction include Mullen scores from the previous two visits. Is this correct?

Response: We are sorry this was not clearer. The binarization of language outcome is based on Mullen expressive and receptive language T scores at the 3-year-old data collection. But the clinical factors to predict the Good/Poor distinction include Mullen ratio scores only from the intake clinical visit (the 1st clinical visit).

The follow-up Mullen scores were not features used to make the outcome predictions. Overall, we used the baseline clinical and MRI measurements collected at intake ages to predict the language outcome measured on average 6 months later.

Why did the authors binarize based on Mullen outcomes? In theory, the authors could use a regression approach rather than classification to predict Mullen scores directly. That would provide more information in the subjects that are on the margin of the Good / Poor threshold.

Response: Thanks for the comment. We followed our previous fMRI work^{3, 4, 10} to consistently binarize language outcome into ASD Good/Poor toddlers in order to examine whether the identified brain structure differences improve performance of the language outcome classification; thus, we tested whether the performance using brain structure in classification is similar to what we discovered from brain activation in response to language task fMRI in the previous work⁴. Moreover, binarizing ASD toddlers into ASD Good/Poor based on language outcome can potentially provide us insights on wisely planning early interventions, where we can distribute the

intervention resources to whom to is most needed (i.e., those with worse language progression: the ASD poor toddlers).

We agree that regression approaches would provide more information for the subjects on the margin of the Good/Poor threshold. We have added this as a future direction in the discussion, where we can train and test more sophisticated regression models by including more brain structure features and exploring different feature reduction techniques.

Did the replication sample have follow up data to split them into Good / Poor to test the generalizability of the SVM classifier.

Response: Unfortunately, we do not have intake and follow-up behavioral (ADOS, Mullen and Vineland assessments) data available for replication sample. However, in the prediction analysis, we have partitioned our samples into training (80% samples) and independent hold-out testing (20% samples) sets, where the training samples were utilized to train and cross-validate the model, and the untouched hold-out testing samples were used to evaluate the generalizability of the SVM classifier.

The authors correct for multiple comparisons over the number of regions analyzed. But they have three different imaging measures for each region (ie SA, volume, thickness). The authors should correct over these too.

Response: Thanks for the comment. We have applied FDR at $p < 0.05$ correction over the three different imaging measures for each region (corrected for 204 comparisons: 68 LH and RH cortical regions, 3 measures (volume/SA/thickness) for each cortical region).

After adding random slope in LME models and applying FDR correction over volume, SA and thickness, group differences of LH caudal middle frontal thickness and LH pericalcarine thickness are not significant, and anterior CC shows significant ASD vs TD differences. Other previously reported regions are still significant. We updated the results and figures in the revised manuscript accordingly. We also redid the language outcome classification analyses based on the updated brain regions and updated the results accordingly.

In figure s5, the authors state that “ASD good toddlers had similar language outcome as TD toddlers”. Did the authors do a direct comparison? If not, it would be interesting to see a ttest comparing scores for ASD good toddlers to TD toddlers.

Response: Thanks for the good suggestion. Since our ASD good toddlers were significantly older than TD toddlers ($p = 5.15E-8$) and female/male ratio was also unmatched between ASD good (10/59) and TD (44/65) toddlers ($p = 2.54E-4$), we tested language outcome differences between ASD good and TD toddlers using an N-way ANOVA test while controlling age, sex, and interaction effects between grouping label (ASD good/TD) and age and sex (i.e., grouping \times age, grouping \times sex). We did not observe significant ASD good vs. TD differences for both Mullen ELT ($p = 0.61$) and RLT ($p = 0.35$) at outcome visit. We have added this test and results in the revised manuscript and supplemental material.

For the classification analyses, it would be helpful for the authors to run multiple iterations of the random splitting into k folds and report the distribution of prediction performances. This will help the reader know the dispersion of the predictions and ensure that any performance is not due to a single “lucky” splitting of the data.

Response: Thank you for the suggestion. For the classification analyses, we have run 100 iterations of 5-fold cross-validation for the Clinic + sMRI model, and computed the classification performance (AUC, accuracy, sensitivity, and specificity) on the held-out testing sample for each iteration. Below figures plot the histogram of resulting AUC, accuracy, sensitivity, and specificity values from held-out testing samples. The frequently observed AUC, accuracy, and sensitivity values are around 80%, 79%, 94%, respectively, which are close to or larger than the reported values (AUC = 79%, accuracy = 79%, sensitivity = 78%). Some iterations even achieve better performance than the reported one. Altogether, the reported performance is unlikely driven by a single “lucky” splitting of the data. We have added this analysis and results in the revised manuscript and supplement (see highlighted).

Reference

1. Shrout PE, Fleiss JL. Intraclass correlations: uses in assessing rater reliability. *Psychological bulletin* **86**, 420 (1979).
2. Koo TK, Li MY. A Guideline of Selecting and Reporting Intraclass Correlation Coefficients for Reliability Research. *J Chiropr Med* **15**, 155-163 (2016).

3. Lombardo MV, *et al.* Atypical genomic cortical patterning in autism with poor early language outcome. *Sci Adv* **7**, (2021).
4. Lombardo MV, *et al.* Different Functional Neural Substrates for Good and Poor Language Outcome in Autism. *Neuron* **86**, 567-577 (2015).
5. Weintraub D, *et al.* Cognition and the course of prodromal Parkinson's disease. *Mov Disord* **32**, 1640-1645 (2017).
6. Grassi M, *et al.* A Novel Ensemble-Based Machine Learning Algorithm to Predict the Conversion From Mild Cognitive Impairment to Alzheimer's Disease Using Socio-Demographic Characteristics, Clinical Information, and Neuropsychological Measures. *Front Neurol* **10**, 756 (2019).
7. Saboo KV, *et al.* Deep learning identifies brain structures that predict cognition and explain heterogeneity in cognitive aging. *Neuroimage* **251**, (2022).
8. Gill S, *et al.* Using Machine Learning to Predict Dementia from Neuropsychiatric Symptom and Neuroimaging Data. *J Alzheimers Dis* **75**, 277-288 (2020).
9. Zielinski BA, *et al.* Longitudinal changes in cortical thickness in autism and typical development. *Brain* **137**, 1799-1812 (2014).
10. Lombardo MV, *et al.* Large-scale associations between the leukocyte transcriptome and BOLD responses to speech differ in autism early language outcome subtypes. *Nature Neuroscience* **21**, 1680-+ (2018).

Reviewers' Comments:

Reviewer #2:

Remarks to the Author:

The authors have adequately addressed the points raised in my initial review and I have no further comments.

Reviewer #3:

Remarks to the Author:

I have one remaining comment that did not come up on my previous review. The authors state: "The sMRI only model leveraged age and sexadjusted intake FreeSurfer measures (age and sex effects were estimated using TD data 104) within regions that showed significant ASD vs. TD differences." Can the authors clarify this a bit? Specifically I am worried about data leakage it appears that regions were select base on the asd vs td differences. While the classification was only in asd, using an asd vs td analysis to inform the classification would break the cross validation (assuming that the same asd subjects were in both the group comparison and the classification). See for example

The effects of data leakage on neuroimaging predictive models

Matthew Rosenblatt, Link Tejavibulya, Rongtao Jiang, Stephanie Noble, Dustin Scheinost
bioRxiv 2023.06.09.544383; doi: <https://doi.org/10.1101/2023.06.09.544383>

Verstynen, T., Kording, K.P. Overfitting to 'predict' suicidal ideation. *Nat Hum Behav* 7, 680–681 (2023). <https://doi.org/10.1038/s41562-023-01560-6>

Reviewer #4:

Remarks to the Author:

In evaluating the manuscript "Language and Social Regions Are Affected in Toddlers with Autism and Predictive of Language Outcome", and in particular the authors' responses to Reviewer 1, I found overall that the authors were sufficiently responsive to most issues raised in the original review. The manuscript reads as stronger given the inclusion of additional details, particularly those related to the analytic strategy and the rationale behind it. I comment on several specific points below.

1. With respect to the authors' response to reviewer #1's first comment, I find the authors' use of "prognostic biomarkers" appropriate given the definition of these terms, even if the window of prediction is rather small. I further agree with the authors' response that the baseline contribution of current language function is expected in the predictive model; however, to reviewer 1's original point, these are the strongest predictors and the MRI data only modestly increase the accuracy. I agree with reviewer 1 that the authors' language around the strength of these findings is a bit overblown (e.g., "great potential as prognostic biomarkers") but am not inclined to quibble about the issue other than to recommend more pragmatic use of language with respect to strength of findings.

2. One issue raised by the original reviewer that I do not believe has been sufficiently addressed and remains a significant weakness. Reviewer #1's second point concerned the use of ADOS scores in the brain X behavior analyses. Their concern was about the comparability of algorithm scores between ADOS versions. They suggested using calibrated severity scores to overcome this limitation.

The authors responded that calibrated severity scores (CSS) are not available for the ADOS-T. However, this is not accurate (see Esler et al. 2015: doi: 10.1007/s10803-015-2432-7). The authors conclude with the claim that the CSS is therefore not "diagnostically, quantitatively or statistically useful". As these scores are available for the toddler module, this is clearly not the case. Moreover, it is questionable to use the raw algorithm scores (as is done in the current

manuscript) given that multiple ADOS modules were used. These scores are not directly comparable as each module contains different items that are scaled and scored differently (e.g., different cut-offs for diagnostic purposes, different score ranges). The authors are further inaccurate in their claim that the CSS does not provide scores for typical toddlers; a CSS of 1-3 indicates low clinical concern and reflects "typical" development with respect to autism symptoms.

3. With respect to reviewer #1's original point seven regarding brain behavior analyses, it is not uncommon in the literature, particularly at this age, to see dissociations between brain X behavior data across groups and thus I understand in part the authors' rationale. However, the reviewer is correct that a unified model accounting for group and group X MRI would account for such dissociations. As is, the authors conducted 624 tests in this portion of the manuscript (13 behaviors X 16 regions X 3 separate models). A single model approach would cut this to 208 analytic models. The authors do not provide a compelling argument for why the single model including an interaction term would not be appropriate given that the interaction term accounts for the very dissociations they have previously observed.

Minor note: I would encourage the authors to select more value neutral designation for the language outcome groups, perhaps "higher vs. lower" rather than "good vs. poor".

Dear Reviewers,

We appreciate your time and effort in helping us to improve the manuscript. We have revised our manuscript, “Language, Social, and Face Regions Are Affected in Toddlers with Autism and Predictive of Language Outcome”, according to your constructive and valuable comments. In the following, we provide detailed responses (in blue color) to each comment. Corresponding changes are highlighted in our revised manuscript.

Reviewer #2 (Remarks to the Author):

The authors have adequately addressed the points raised in my initial review and I have no further comments.

Response: We thank the Reviewer for being satisfied with our responses and confirming no further comments.

Reviewer #3 (Remarks to the Author):

I have one remaining comment that did not come up on my previous review. The authors state: "The sMRI only model leveraged age and sex-adjusted intake FreeSurfer measures (age and sex effects were estimated using TD data) within regions that showed significant ASD vs. TD differences." Can the authors clarify this a bit? Specifically I am worried about data leakage it appears that regions were select base on the asd vs td differences. While the classification was only in asd, using an asd vs td analysis to inform the classification would break the cross validation (assuming that the same asd subjects were in both the group comparison and the classification). See for example

*The effects of data leakage on neuroimaging predictive models
Matthew Rosenblatt, Link Tejavibulya, Rongtao Jiang, Stephanie Noble, Dustin Scheinost
bioRxiv 2023.06.09.544383; doi: <https://doi.org/10.1101/2023.06.09.544383>*

Verstynen, T., Kording, K.P. Overfitting to ‘predict’ suicidal ideation. Nat Hum Behav 7, 680–681 (2023). <https://doi.org/10.1038/s41562-023-01560-6>

Response: We thank the Reviewer for this insightful comment. For the sMRI only model in the ASD language outcome prediction, we focused on the Aim 1-identified 16 robust and replicable brain regions showing significant ASD vs TD difference. The reason is that the language outcome prediction was designed as a follow up analysis to exam whether these ASD discriminating brain regions have any clinical indications, especially for the prediction of language that is one of the key outcomes for toddlers with ASD.

The phenotype of interest for feature selection (ASD vs TD differences) and phenotype of interest for prediction (ASD toddlers with good/poor language outcome) were NOT the same, although 157 out of 166 ASD toddlers used for examining ASD vs TD difference were included for ASD language outcome prediction. I further examined ASD poor vs ASD good differences of all Freesurfer measurements in the *entire* ASD dataset that were used for language outcome prediction. Among top 16 regions that are promising for predicting ASD poor/good language outcome, only

the volume of mid anterior CC overlapped with the 16 regions showing robust ASD vs TD differences. Moreover, the volume of mid anterior CC is one of the top 16 regions showing promising ASD good vs ASD poor differences in the 124 *training* subjects. Although the volume of mid anterior CC had moderate contribution to language outcome prediction, other 9 brain regions also show moderate to large contributions (Figure S7). Altogether, feature/data leakage was not the case for the current analysis and the prediction accuracy from neuroimaging measurements was unlikely to be inflated.

We agree that a nested K-fold cross validation to select top features from all brain regions parcellated from FreeSurfer will be a rigorous experiment design for evaluating how much neuroimaging measurements can predict language outcome, but this analysis will make the current work lack of focus especially when the top brain features for language outcome prediction were different from those brain regions showing ASD vs TD differences. The current work focused on identifying replicable brain regions showing ASD vs TD differences and evaluating their clinical translational value in terms of language outcome prediction for ASD toddlers. We included the language prediction analysis using a nested K-fold cross validation to select top features as one of the feature directions in the revised Discussion (see highlighted).

Reviewer #4 (Remarks to the Author):

In evaluating the manuscript “Language and Social Regions Are Affected in Toddlers with Autism and Predictive of Language Outcome”, and in particular the authors’ responses to Reviewer 1, I found overall that the authors were sufficiently responsive to most issues raised in the original review. The manuscript reads as stronger given the inclusion of additional details, particularly those related to the analytic strategy and the rationale behind it. I comment on several specific points below.

Response: Thanks for your comments.

1. With respect to the authors' response to reviewer #1's first comment, I find the authors' use of "prognostic biomarkers" appropriate given the definition of these terms, even if the window of prediction is rather small. I further agree with the authors' response that the baseline contribution of current language function is expected in the predictive model; however, to reviewer 1's original point, these are the strongest predictors and the MRI data only modestly increase the accuracy. I agree with reviewer 1 that the authors' language around the strength of these findings is a bit overblown (e.g., "great potential as prognostic biomarkers") but am not inclined to quibble about the issue other than to recommend more pragmatic use of language with respect to strength of findings.

Response: We thank the Reviewer for this suggestion and have removed such wording.

2. One issue raised by the original reviewer that I do not believe has been sufficiently addressed and remains a significant weakness. Reviewer #1's second point concerned the use of ADOS scores in the brain X behavior analyses. Their concern was about the comparability of algorithm scores between ADOS versions. They suggested using calibrated severity scores to overcome this limitation.

The authors responded that calibrated severity scores (CSS) are not available for the ADOS-T. However, this is not accurate (see Esler et al. 2015: doi: 10.1007/s10803-015-2432-7). The authors conclude with the claim that the CSS is therefore not “diagnostically, quantitatively or statistically useful”. As these scores are available for the toddler module, this is clearly not the case. Moreover, it is questionable to use the raw algorithm scores (as is done in the current manuscript) given that multiple ADOS modules were used. These scores are not directly comparable as each module contains different items that are scaled and scored differently (e.g., different cut-offs for diagnostic purposes, different score ranges). The authors are further inaccurate in their claim that the CSS does not provide scores for typical toddlers; a CSS of 1-3 indicates low clinical concern and reflects “typical” development with respect to autism symptoms.

Response: We apologize for our unclear response regarding the ADOS CSS score. ADOS data for this study were collected from June 2007-February 2013 using the original version of the ADOS, and 163 out of 275 subjects included in this study received the Toddler module. The newest ADOS version is the ADOS-2 which was released in 2012, and most of the published CSS validation work has been done using data collected using that new ADOS-2 version. Note that within the newest ADOS-2 version, CSS scores for Modules 1-3 are printed by WPS directly on the back of the protocol booklet for regular use, whereas for the Toddler module (and Module 4), CSS scores are not printed on the protocol booklet. Moreover, although the initial work on the CSS score started with Gotham’s 2009 publication¹ which utilized the original ADOS consistent with the time frame of our ADOS data collection, this Gotham paper did not contain CSS scores for the Toddler Module either. Thus, at this point, we do not believe that Toddler Module CSS scores are available for the original ADOS version.

There are also reasons why using the Total score has some advantages over the CSS for our present study. Although we agree that the CSS scores make it easier to compare across modules, the primary reason Lord and colleagues developed the CSS was to create a metric that is relatively independent of participant characteristics such as language ability and to create a metric suitable for tracking changes across development. Our study does not examine change across development and instead highlights the temporal lobe, a brain region key in both language and social processing- in essence social communication. As such, we use metrics with language relevance, namely, the ADOS total score. In fact, the initial study by Gotham showed that verbal IQ accounts for 43% of the variance in ADOS raw total score, but only 10% of the CSS¹.

We do, however, understand the Reviewer’s perspective, and acknowledge this as a limitation in the Discussion. Specifically, we note that there can be minor differences in Total Scores across the modules in the Discussion.

3. With respect to reviewer #1's original point seven regarding brain behavior analyses, it is not uncommon in the literature, particularly at this age, to see dissociations between brain X behavior data across groups and thus I understand in part the authors' rationale. However, the reviewer is correct that a unified model accounting for group and group X MRI would account for such dissociations. As is, the authors conducted 624 tests in this portion of the manuscript (13 behaviors X 16 regions X 3 separate models). A single model approach would cut this to 208 analytic models. The authors do not provide a compelling argument for why the single model including an

interaction term would not be appropriate given that the interaction term accounts for the very dissociations they have previously observed.

Response: We are sorry we were not clear enough about the three major aims of our study. We have changed the Introduction adding information and clarifying the three study aims. The first aim was to identify replicable MRI biomarkers of ASD at intake ages. To achieve this purpose, we comprehensively analyzed MRI regions and measures (volume, thickness, surface area) to identify MRI biomarkers that were differentially enlarged or reduced relative to the typical brain with factoring out the overall brain size.

With this knowledge of MRI biomarkers of ASD, the second and third aims could be achieved. These were to statistically test whether these MRI biomarkers provide two important types of clinical translational information at the early-age of first clinical presentation: namely ASD prognostic information and information about ASD toddler's symptom and language severity.

KEY POINT: To achieve these two aims, it is completely unnecessary to include typical toddlers, typical development, or the TD study group. We are *not* trying to predict the typical child's clinical outcome using MRI; that would be clinically pointless. We are *not* trying index a child's social severity or impaired language development using MRI; that would be clinically pointless too. This is why the language outcome prediction analysis does *not* include TD but only ASD subgroups. Likewise, this is why the brain-behavior analysis does *not* include TD but only ASD.

ANOTHER KEY POINT: While we originally included TD in the brain-behavior section as a separate analysis that we thought might be of interest to some child development scientists, we now realize this tangential information is an unnecessary confusion, and we are sorry that happened. To avoid this, we now remove TD analyses from main text in the paper.

Minor note: I would encourage the authors to select more value neutral designation for the language outcome groups, perhaps "higher vs. lower" rather than "good vs. poor".

Response: We agree that good and poor are not sufficiently value neutral. Higher and lower terms also carry some degree of "better and not so good", which are also not entirely value neutral. Therefore, in our revision, we have chosen designations "pre-profound autism and strong autism" that reflect current clinical subtype relevance and meaning. The term 'profound autism' was first introduced 2 years ago by Cathy Lord and colleagues in the *Lancet Commission on the Future of Care and Clinical Research in Autism* ². One of the many points articulated by this paper have to do with challenges associated with providing optimal clinical care (and conducting research) within the spectrum of ASD. The paper argues that ASD is a broad term that contains individuals with severe cognitive, language, and social impairments, estimated by the commission to be about 50% of the ASD population ², as well as those who are more able (have "stronger" abilities)- and yet, all have the same "ASD" diagnostic label. Only a few papers have been published since the Lancet Commission, including a new report by the CDC that estimates approximately 27% of ASD individuals have "profound" autism ³. The impact of the use of the term is not without debate⁴. Although the precise definition and criteria of "profound" are not yet fully agreed upon in the field, what is clear is that the field sees the need to understand the clinical and biological differences among different ASD subtypes and predicting at the earliest ages who might need life-long care, can have considerable advantages for family and society alike. Our manuscript takes the important step in recognizing this issue and reporting the first attempt to use a combination of biological and

clinical variables to objectively separate those who may have a strong outcome from those pre-profound toddlers who will eventually have a “profound” autism outcome.

Reference

1. Gotham K, Pickles A, Lord C. Standardizing ADOS scores for a measure of severity in autism spectrum disorders. *J Autism Dev Disord* **39**, 693-705 (2009).
2. Lord C, *et al.* The Lancet Commission on the future of care and clinical research in autism. *Lancet* **399**, 271-334 (2022).
3. Hughes MM, *et al.* The Prevalence and Characteristics of Children With Profound Autism, 15 Sites, United States, 2000-2016. *Public Health Rep* **138**, 971-980 (2023).
4. Kripke-Ludwig R. “Profound Autism” Label Does Not Predict Strengths or Help Plan Supports. *Public Health Reports* **138**, 849-850 (2023).

Reviewers' Comments:

Reviewer #3:

None

Reviewer #5:

Remarks to the Author:

I appreciate the opportunity to review this manuscript outlining differences in regional brain volume, surface area, and thickness and their links to cognition/behavior in young toddlers with ASD. Here I focus mainly on response to points raised by prior reviewers and a few minor additional suggested edits.

I agree with the author's rationale for excluding the TD brain-behavior analyses from the manuscript and placing them in the supplement. However, I would suggest adding in a sentence to the results section of the main text that refers readers to the supplement for that information (as it reads, I missed that this was in the supplement). The authors may also consider including a sentence in the discussion interpreting the findings across groups (e.g., findings were not consistent between groups suggesting that the neurobiology of social communication in ASD may differ from that observed in TD).

Author's use of "pre-profound ASD" and "strong ASD" was rooted in an attempt to be thoughtful about language, but this terminology is confusing. "Strong ASD" is not a common term in the literature that may in fact be misinterpreted to suggest a more "severe" presentation of symptoms at first glance, which is the opposite of the intention here. Similarly, "pre-profound" is not clear, and not wholly in line with the classifications set forth by Lord and colleagues. I suggest the authors stick with objective terms like "Low" for $T < 40$ and "Low/Average" $T \geq 40$ to refer to where a child scores on the ranking of the Mullen T scores.

With regards to the points raised in prior reviews about the ADOS calibrated severity scores, I take the authors point and think their use of total scores is defensible.

Other suggested edits:

1. Introduction line 48: "Some toddlers may earn college degrees" should be "Some toddlers may go on to earn college degrees".
2. Introduction sentence spanning lines 56-59 needs a reference for the described cortical organoid linkages to behavior in ASD.
3. Introduction sentence spanning lines 83-85 is an overstatement. See Hazlett et al., 2017.
4. TABLE 1: Consider adding a footnote describing the calculation of the ratio scores since these are not standard.
5. Supplement: Figure S3 legend should include reference to color of groups (e.g., what is red vs. blue).

Dear Reviewers,

We appreciate your time and effort in helping us to improve the manuscript. We have revised our manuscript entitled “Differences in regional brain structure in toddlers with autism are related to future language outcomes” (old title: “Language, Social and Face Regions Are Affected in Toddlers with Autism and Predictive of Language Outcome”), according to your constructive and valuable comments. In the following, we provide detailed responses (in blue color) to each comment. Corresponding changes are highlighted in our revised manuscript.

Reviewer #5 (Remarks to the Author):

I appreciate the opportunity to review this manuscript outlining differences in regional brain volume, surface area, and thickness and their links to cognition/behavior in young toddlers with ASD. Here I focus mainly on response to points raised by prior reviewers and a few minor additional suggested edits.

I agree with the author’s rationale for excluding the TD brain-behavior analyses from the manuscript and placing them in the supplement. However, I would suggest adding in a sentence to the results section of the main text that refers readers to the supplement for that information (as it reads, I missed that this was in the supplement). The authors may also consider including a sentence in the discussion interpreting the findings across groups (e.g., findings were not consistent between groups suggesting that the neurobiology of social communication in ASD may differ from that observed in TD).

Response: Thanks for the suggestion. We have briefly added TD brain-behavior association results in the Results section and refer readers to the supplement for full information (see highlighted). We also added the suggested sentence in the discussion to interpret the findings across groups (see highlighted).

Author’s use of “pre-profound ASD” and “strong ASD” was rooted in an attempt to be thoughtful about language, but this terminology is confusing. “Strong ASD” is not a common term in the literature that may in fact be misinterpreted to suggest a more “severe” presentation of symptoms at first glance, which is the opposite of the intention here. Similarly, “pre-profound” is not clear, and not wholly in line with the classifications set forth by Lord and colleagues. I suggest the authors stick with objective terms like “Low” for $T < 40$ and “Low/Average” $T \geq 40$ to refer to where a child scores on the ranking of the Mullen T scores.

Response: Thanks for the suggestion! We changed the language outcome grouping to the suggested objective terms: we use “Low” to refer a child’s Mullen expressive language and receptive language T scores are less than 40, and “Low/Average” to refer a child’s Mullen expressive language or receptive language T score is equal to or larger than 40.

With regards to the points raised in prior reviews about the ADOS calibrated severity scores, I take the authors point and think their use of total scores is defensible.

Response: We thank the Reviewer for being satisfied with our responses regarding ADOS calibrated severity scores.

Other suggested edits:

1. Introduction line 48: “Some toddlers may earn college degrees” should be “Some toddlers may go on to earn college degrees”.

Response: Thanks for the comment. We changed it to “Some toddlers may go on to earn college degrees”.

2. Introduction sentence spanning lines 56-59 needs a reference for the described cortical organoid linkages to behavior in ASD.

Response: Thanks for the comment. Introduction sentence spanning lines 56-59 were summarized from our recent manuscript under revision in Molecular Psychiatry. We have added the corresponding reference.

3. Introduction sentence spanning lines 83-85 is an overstatement. See Hazlett et al., 2017.

Response: Thanks for the comment. We have corrected Introduction lines 83 to 85 accordingly (see highlighted).

4. TABLE 1: Consider adding a footnote describing the calculation of the ratio scores since these are not standard.

Response: Thanks for the comment. We have added a footnote to describe the calculation of the ratio scores.

5. Supplement: Figure S3 legend should include reference to color of groups (e.g., what is red vs. blue).

Response: Thanks for the comment. Dots with red color indicate ASD toddlers and dots with blue color indicate TD toddlers. We have added this information in the legend of Figure S6 (old numbering corresponds to Figure S3).